# PCPE-1, a brown adipose tissue-derived cytokine, promotes obesity-induced liver fibrosis

Yung Ting Hsiao[1], Yohko Yoshida[2,3], Shujiro Okuda[4], Manabu Abe [5,6], Seiya Mizuno [7], Satoru Takahashi [7], Hironori Nakagami [8], Ryuichi Morishita[8], Kenya Kamimura [9,10], Shuji Terai [9], Tin May Aung [1], Ji Li[11], Takaaki Furihata[2], Jing Yuan Tang[2], Kenneth Walsh[12], Akihito Ishigami [13], Tohru Minamino [2] & Ippei Shimizu [1,14]✉

## Abstract

**Metabolic dysfunction-associated steatohepatitis (MASH, previously termed non-alcoholic steatohepatitis (NASH)), is a major complication of obesity that promotes fatty liver disease. MASH is characterized by progressive tissue fibrosis and sterile liver inflammation that can lead to liver cirrhosis, cancer, and death. The molecular mechanisms of fibrosis in MASH and its systemic control remain poorly understood. Here, we identified the secreted-type pro-fibrotic protein, procollagen C-endopeptidase enhancer-1 (PCPE-1), as a brown adipose tissue (BAT)-derived adipokine that promotes liver fibrosis in a murine obesity-induced MASH model. BAT-specific or systemic PCPE-1 depletion in mice ameliorated liver fibrosis, whereas, PCPE-1 gain of function in BAT enhanced hepatic fibrosis. High-calorie diet-induced ER stress increased PCPE-1 production in BAT through the activation of IRE-1/JNK/c-Fos/c-Jun signaling. Circulating PCPE-1 levels are increased in the plasma of MASH patients, suggesting a therapeutic possibility. In sum, our results uncover PCPE-1 as a novel systemic control factor of liver fibrosis.**

**Keywords** BATokine; PCPE-1; Fibrosis; Obesity; MASH
**Subject Categories** Metabolism; Molecular Biology of Disease

## Introduction

Excessive calorie intake induces chronic sterile inflammation in visceral adipose tissue. An inflamed visceral fat pad produces pro-inflammatory adipokines including interleukin-1β (IL-1β), interleukin-6 (IL-6), and tumor necrosis factor-α (TNF-α) (Ouchi et al, 2011; Rosen and Spiegelman, 2014), which contribute to the progression of systemic insulin resistance, type 2 diabetes, MAFLD, heart failure, and atherosclerosis (Gehrke and Schattenberg, 2020; Mantovani et al, 2022; Stefan and Cusi, 2022). These adipokines were tested for their therapeutic potential in clinical trials with their respective neutralizing antibodies, with varying outcomes depending on the disease or target (Mann et al, 2004; Ridker et al, 2017). Exploring druggable adipokines continue to be a realistic and interesting approach to developing next-generation therapies for cardiovascular-metabolic diseases. In contrast to adipokines produced from white adipose tissue (WAT), BATokines have received only limited attention in the literature. BAT was initially characterized as a thermogenic organ, but has now come to be recognized as having crucial roles in the maintenance of systemic metabolism (Bartelt et al, 2011; Stanford et al, 2013; Yoneshiro et al, 2013). Indeed, BAT has been assessed for its endocrine function, as well as for its metabolic characteristics, in recent years. For example, BAT-derived neuregulin 4 was reported to alleviate endothelial inflammation and atherosclerosis (Shi et al, 2022). Ependymin-related protein 1 (EPDR1) has also been identified as a secreted-type BATokine involved in the enhancement of pancreatic β-cell function (Cataldo et al, 2022). Given the reported protective effects of BAT-derived adipokines, the question remains: Does BAT also produce pathogenic adipokines? We have previously showed that BAT becomes hypoxic and exhibits a functional decline in obesity due to a reduction in vascular endothelial growth factor a

[1]Department of Cardiovascular Aging, National Cerebral and Cardiovascular Center Research Institute, Osaka 564-8565, Japan. [2]Department of Cardiovascular Biology and Medicine, Juntendo University Graduate School of Medicine, Tokyo 113-8421, Japan. [3]Department of Advanced Senotherapeutics, Juntendo University Graduate School of Medicine, Tokyo 113-8421, Japan. [4]Division of Bioinformatics, Niigata University Graduate School of Medical and Dental Sciences, Niigata 951-8510, Japan. [5]Department of Cellular Neurobiology, Brain Research Institute, Niigata University, Niigata 951-8585, Japan. [6]Department of Animal Model Development, Brain Research Institute, Niigata University, Niigata 951-8585, Japan. [7]Laboratory Animal Resource Center in Transborder Medical Research Center, Institute of Medicine, University of Tsukuba, Ibaraki 305-8577, Japan. [8]Department of Health Development and Medicine, Osaka University Graduate School of Medicine, Osaka 565-0871, Japan. [9]Division of Gastroenterology and Hepatology, Graduate School of Medical and Dental Sciences, Niigata University, Niigata 951-8510, Japan. [10]Department of General Medicine, Niigata University School of Medicine, Niigata 951-8510, Japan. [11]Department of Cardiology, 2nd Affiliated Hospital of Harbin Medical University, Harbin 150001, PR China. [12]Division of Cardiovascular Medicine, Robert M. Berne Cardiovascular Research Center, University of Virginia School of Medicine, Charlottesville, VA 22908, USA. [13]Molecular Regulation of Aging, Tokyo Metropolitan Institute for Geriatrics and Gerontology, Tokyo 173-0015, Japan. [14]Department of Cardiovascular Medicine, National Cerebral and Cardiovascular Center, Osaka 564-8565, Japan. ✉E-mail: shimizu.ippei@ncvc.go.jp

(VEGF-A) levels in this organ (Shimizu et al, 2014). We thus hypothesized that BAT-derived adipokines could be further explored through analysis of transcriptome data on BAT from a high-fat diet (HFD)-fed obese mice or adipose tissue-specific *Vegfa* knockout mice (Adipo-*Vegfa* KO). Using a bioinformatic approach and drawing on the analysis of a murine BAT-specific knockout model, we found that PCPE-1 could be classified as a BATokine involved in the progression of fibrosis in the liver. As many as 25.2% of the global population is considered to have MAFLD (Younossi et al, 2016), a progressive disease that transitions to a more severe form known as MASH (Sheedfar et al, 2013), which is characterized by liver fibrosis, and chronic inflammation and may progress to cirrhosis and hepatocellular carcinoma (Sheedfar et al, 2013). Many unmet medical needs exist, particularly for MASH, among obesity-related fibrotic diseases, given that the mechanisms of liver fibrosis in MASH are not fully elucidated, and the therapeutic options for this slowly progressive and deadly disorder remain rather limited (Harrison et al, 2023; Sheedfar et al, 2013; Takaki et al, 2013). As MASH is expected to supercede hepatitis B or C as the leading cause of cirrhosis, it is becoming urgent to explore druggable targets for this critical disorder (Lazarus et al, 2022). Of these, PCPE-1 (encoded by the *Pcolce* gene) is of interest as a secreted-type, pro-fibrotic molecule that binds to the C-terminal propeptide of type I procollagen thereby enhancing the activity of procollagen C-proteinase including bone morphogenetic protein 1 (BMP-1). PCPE-1 has been shown to accelerate procollagen maturation through cleavage of the C-propeptide of pro-collagen and to have a critical role in the production of mature and structured collagen fibrils (Lagoutte et al, 2021). However, the mechanisms involved in the production of this secreted, profibrotic protein, i.e., an initiator and enhancer of tissue fibrosis, remain unclear. Here we present evidence that endoplasmic reticulum (ER) stress-mediated signaling increases PCPE-1 production in BAT thus contributing to the progression of liver fibrosis in the murine obese-MASH model.

## Results

### Exploration of BAT-derived adipokine in obesity

To assess BATokines expression under pathogenic conditions, two DNA microarray datasets obtained from HFD-fed obese mice (Gene Expression Omnibus: GSE28440), or Adipo-*Vegfa* KO mice (GSE221854) were examined. This analysis demonstrated that the relative expression of the transcript *Pcolce*, the encoding gene for PCPE-1, was highest in the BAT of adipo-*Vegfa* KO mice compared to that of littermate control mice (Fig. 1A). Analysis of GSE28440 (Fitzgibbons et al, 2011a, b) also showed that *Pcolce* increased in the BAT of dietary obese mice compared to that of mice fed a normal chow (Fig. 1A). To confirm this finding, diet-induced obese mice were generated by maintaining them on a high-fat diet (HFD). It was found that, in addition to body weight, BAT weight increased (Fig. EV1A,B) as previously reported (Shimizu et al, 2014). These HFD-fed mice were shown to develop a whitened phenotype of BAT with a quantitative PCR study showing that the transcript *Pcolce* was predominantly expressed in BAT (Fig. 1B,C). Analysis of the Reference Expression Dataset (RefEx) (Ono et al, 2017) showed that the transcript *Pcolce* was most highly expressed in adipose

tissue, followed by the pineal gland, bone, muscle, and adrenal gland (Fig. EV1C). Analysis of GSE8044 (Seale et al, 2007a; Seale et al, 2007b) showed that the transcript *Pcolce* was more highly expressed in BAT than in epididymal WAT (eWAT) (Fig. EV1D). GSE64718 (Kim et al, 2015a, b), GSE123394 (Almind and Kahn, 2004a, b), GSE194075 (Ali et al, 2022a, b), GSE216327 (Witt et al, 2023a, b), GSE96932 (Ridaura et al, 2018a, b), and GSE171710 (Mia et al, 2021a, b) were also tested for *Pcolce* expression in obesity in BAT, eWAT, liver, skeletal muscle (SM), bone, adrenal gland, and heart. It was found that *Pcolce* was increased significantly in BAT (Fig. EV1E), and significantly increased, albeit to a much lesser degree, in the liver (Fig. EV1F). However, it was not significantly different in the other organs (eWAT, bone, adrenal gland, skin, and heart) or reduction (SM) (Fig. EV1F–J). Next, analysis of single-cell RNA-seq data deposited in the Tabula Muris Senis (Tabula Muris, 2020) demonstrated that *Pcolce* was highly expressed in BAT, WAT, aorta, and heart (Fig. EV1K), where *Pcolce* was most highly expressed in the fibroblasts in the heart (Fig. EV1L) as well as in the mesenchymal stem cells of adipose tissue in BAT (Fig. EV1M). Furthermore, analysis of cardiac fibroblasts and brown adipocytes from neonatal rats and subjected to cell culture showed that *Pcolce* was most highly expressed in brown adipocytes (Fig. EV1N). Analysis of hepatocytes and brown adipocytes extracted from adult male C57BL/6NCrSlc mice also showed that *Pcolce* expression was much higher in brown adipocytes both in primary cells and cell lines (Fig. EV1O,P). It was also found that the protein PCPE-1 was more highly expressed in BAT or in the circulation in dietary obesity (Fig. 1D–F). Consistent with the pathological findings on MASH, it was found that dietary obese mice had developed liver steatosis (Fig. 1G,H), characterized by an increase in their circulatory alanine aminotransferase (ALT) levels (Fig. 1I), liver fibrosis (Figs. 1J,K,L and EV1Q) and inflammatory markers (Fig. 1L), which were also associated with an increase in their PCPE-1 levels in the liver (Fig. 1M). While one of the challenging clinical issues in MASH arises from lack of biomarkers (Ramai et al, 2021), our studies of human samples showed that the circulatory levels of PCPE-1 increased in patients with MASH in both BMI-adjusted (BMI < 25) and non-BMI-adjusted groups (Fig. 1N; Appendix Fig. S1A,B, Table 1), thus suggesting a role for PCPE-1 as a biomarker for this condition.

### BATokine PCPE-1 promotes liver fibrosis in MASH

To further test the role of BAT-derived PCPE-1, a BAT-specific PCPE-1 knockout model was generated by crossing Ucp1-Cre[+/-] with *Pcolce*[flox/flox] mice (BAT *Pcolce* KO) (Fig. EV2A). In this model, the expression of the transcript *Pcolce* was shown to be reduced in BAT but was comparable in the liver, epididymal WAT, heart, and skeletal muscle (Figs. 2A and EV2B). It was also found that BAT *Pcolce* KO mice exhibited reduced PCPE-1 expression in BAT, plasma, and liver in dietary obesity (Figs. 2B–D and EV2C). In this setting, it was found that body weight, BAT and WAT weights, and pathological findings on BAT were comparable between the genotypes compared (Figs. EV2D–F; Appendix Figs. S2A,S2B), while liver fibrosis was ameliorated in BAT *Pcolce* KO mice compared to littermate control mice in our obese-MASH model (Figs. 2E,F and EV2G,H). It was also found that liver triglyceride (TG) was reduced in the BAT *Pcolce* KO model (Fig. 2G), while the levels of plasma ALT (Fig. 2H), as well as levels of expression of the

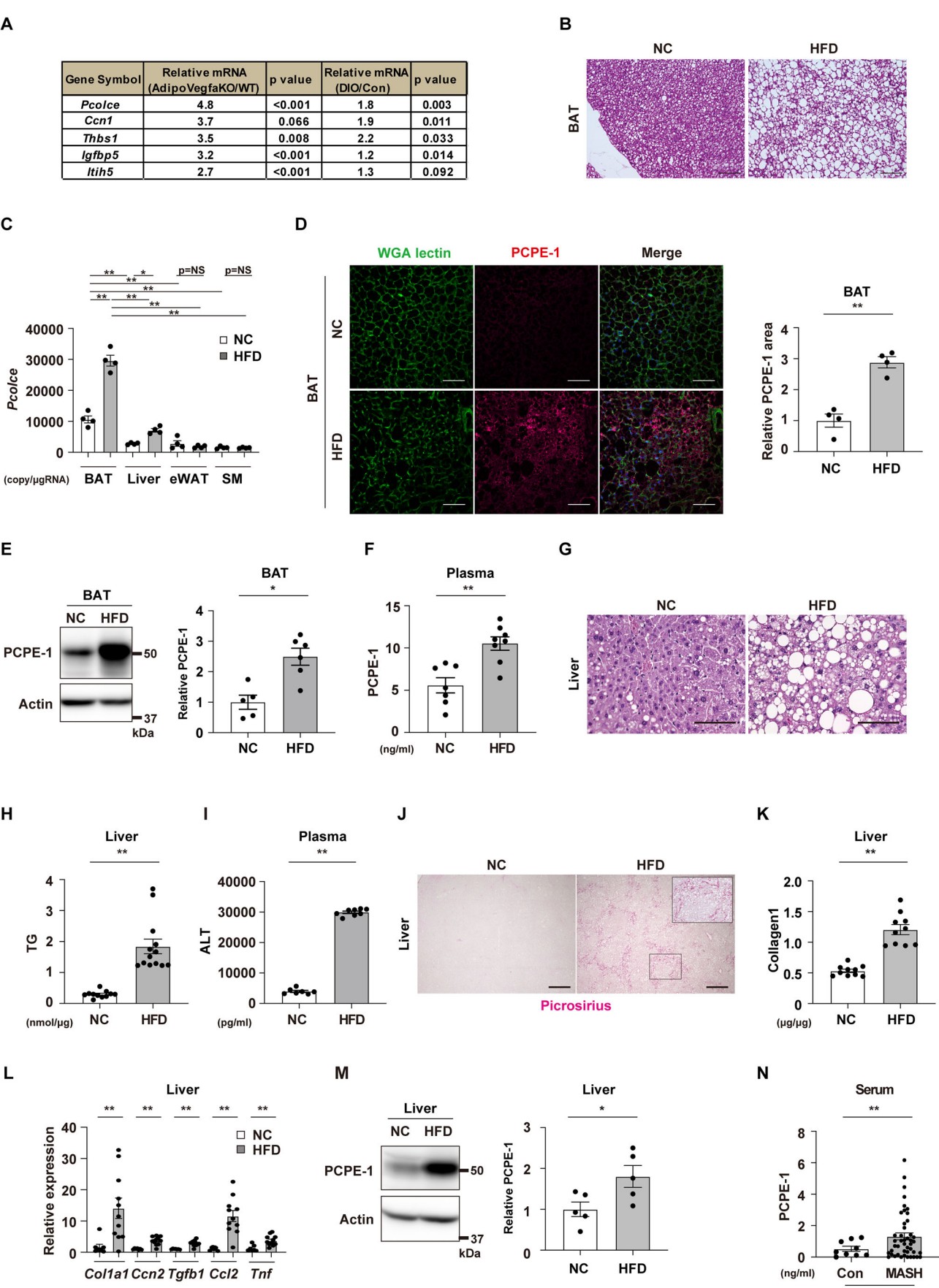

**Figure 1. Exploration of BAT-derived adipokine in obesity.**

(A) Results of DNA microarray datasets analyzing brown adipose tissue (BAT) from adipose tissue-specific *Vegfa* knockout mice (Adipo-*Vegfa* KO) (GSE221854) or high-fat diet (HFD) fed wild-type mice (diet-induced obesity (DIO)) (GSE28440). (B) C57BL/6NCrSlc mice were fed with a normal chow (NC) or a high-fat diet (HFD) from 4 weeks of age and studied at 34–38 weeks of age. Hematoxylin-eosin (HE) staining of BAT from indicated mice. Scale bar = 100 μm. (C) Copy number for the transcript *Pcolce* in BAT, liver, epididymal white adipose tissue (eWAT) and skeletal muscle (SM) (quadriceps) of the indicated mice (all $n = 4$). (D) Immunofluorescent staining for procollagen C-endopeptidase enhancer-1 (PCPE-1) in BAT of the indicated mice. Scale bar = 50 μm. The right panel indicates the quantification of the PCPE-1 positive area in BAT ($n = 4, 4$). (E) Western blot analysis of PCPE-1 expression in BAT from the indicated mice. The right panel indicates the quantification of PCPE-1 relative to Actin ($n = 5, 6$). (F) ELISA for PCPE-1 in plasma from the indicated mice ($n = 7, 8$). (G) HE staining of the liver from the indicated mice. Scale bar = 100 μm. (H, I) ELISA for liver triglyceride (TG) ($n = 11, 13$) (H) or plasma alanine transaminase (ALT) ($n = 7, 8$) (I) in the indicated mice. (J) Picrosirius red staining of liver from the indicated mice. Scale bar = 500 μm. The quantification results see Fig. EV1Q. (K) ELISA for liver collagen type I (Collagen1) in the indicated mice ($n = 10, 10$). (L) Results from quantitative PCR (qPCR) showing transcripts *Col1a1* ($n = 8, 11$), *Ccn2*(Ctgf) ($n = 7, 11$), *Tgfb1* ($n = 7, 11$), *Ccl2* ($n = 7, 11$), and *Tnf* ($n = 7, 11$) in the liver from the indicated mice. (M) Western blot analysis of PCPE-1 expression in liver from the indicated mice. The right panel indicates the quantification of PCPE-1 relative to Actin ($n = 5, 5$). (N) ELISA for human PCPE-1 in serum from control (Con) or MASH patients ($n = 10, 45$). Except for (C), all data were analyzed by an independent-samples T-test. The data in (C) were analyzed by a two-way analysis of variance (ANOVA) followed by Tukey's multiple comparison test. All data are from different biological replicates. In (N), analyses were performed including values categorized as outliers. Data information: Representative data of two independent series (C–E, L, M), one independent experiment analyzing samples from at least 2 independently prepared samples (F, H, I, K, N). *$p < 0.05$, **$p < 0.01$. Values represent the mean ± SEM. NS = not significant. Source data are available online for this figure.

**Table 1. Characterization of patients' background.**

|  | Con ($n = 10$) | NASH ($n = 45$) | *p* value |
|---|---|---|---|
| Age | 56.1 ± 13.9 | 59.1 ± 18.6 | NS |
| Female | 6/10 (60%) | 23/45 (51%) | NS |
| BMI | 22.1 ± 2.0 | 25.9 ± 4.3 | $p < 0.001$ |

Data for age, gender, and BMI are described. The control (non-MASH) group included patients with the following diagnosis; Paroxysmal supraventricular tachycardia (PSVT) ($n = 5$), Premature ventricular conduction (PVC) ($n = 3$), Vasospastic angina (VSA) ($n = 2$). Values represent the mean ± SD. Data were analyzed by an independent-samples T-test (Age and BMI) or by a Fisher's exact test. Source data is available online for this table.

transcripts for fibrillogenesis, were similar between the genotypes compared except for *Tnf*, which showed a reduction in BAT *Pcolce* KO mice (Fig. 2I).

Next, analysis of a BAT-specific PCPE-1 gain of function model was generated by incorporating adeno-associated virus (AAV) encoding for the mouse transcript *Pcolce* (AAV-*mPcolce* model) (Fig. 2J–L). In this model, the transcript *Pcolce* was not shown to be increased in heart, liver, and kidney (Fig. EV2I), while levels of PCPE-1 in circulation, as well as fibrosis in the liver, were shown to be increased in mice fed a normal chow-fed (Figs. 2M,N,O and EV2J). The levels of liver TG (Fig. 2P) and plasma ALT (Fig. 2Q), as well as the levels of expression of transcripts for fibrosis or inflammation in the liver, were similar between the groups compared (Fig. 2R). AAV-mPcolce induction led to slight reductions in body weight, but there was no change in the weight and pathological findings of BAT, which were found comparable between the groups (Fig. EV2K–M, Appendix Fig. S2C). Further-more, analysis of a AAV-mPcolce model generated in the dietary obesity group revealed similar hematoxylin-eosin (HE) stain findings in BAT between the groups compared (Appendix Fig. S2D). Again, the expression profiles of transcripts for fibrillogenesis markers and inflammation were shown to be comparable between the groups (Fig. EV2N). An examination of BAT of HFD-fed obese mice in GSE28440 revealed an increase in collagen-related markers including *Adamts2* and *Ccn2* (Ctgf) (Appendix Fig. S2E) in addition to *Ccn1* (Cyr61) (Fig. 1A), while these markers were shown to be similar in the AAV model on HFD or normal diet

(Appendix Fig. S2F), as well as in BAT *Pcolce* KO mice and their control mice fed a HFD (Appendix Fig. S2G). Thus, it was found that AAV treatment increased circulatory and liver levels of PCPE-1 under these conditions (Fig. EV2O,P), while HFD led to an increase in liver fibrosis, and *Pcolce* induction in BAT synergistically augmented liver fibrosis (Fig. EV2Q; Appendix Fig. S2H and S2I). These findings suggest that PCPE-1 was predominantly produced from BAT and promoted liver fibrosis. In the liver of the AAV model with dietary obesity, it was found that TG levels were significantly increased in the AAV-mPcolce group (Fig. EV2R), while plasma ALT levels were similar between the groups (Fig. EV2S). Treatment of hepatocytes with a recombinant PCPE-1 protein produced no change in the level of transcripts involved in the uptake, synthesis, lipolysis, or metabolism of lipids (Appendix Figs. S3A–D). Furthermore, it was shown that intracellular TG was increased in steatotic hepatocytes, but with this change not affected by PCPE-1 overexpression (Appendix Fig. S3E).

## Models with systemic PCPE-1 inhibition ameliorate liver fibrosis

In an analysis of the systemic *Pcolce* KO model (*Pcolce* KO) maintained on a chow or a HFD, which was conducted to further confirm the therapeutic potential of PCPE-1, it was shown that, in addition to BAT, transcript *Pcolce* was reduced in the liver, epididymal WAT, heart, and skeletal muscle in the KO mice (Figs. 3A and EV3A). It was also found that the PCPE-1 protein was reduced in BAT and circulation in KO mice (Figs. 3B,C and EV3B), while no changes were observed in body weight and BAT weight between the genotypes while on their respective diets (Fig. EV3C,D). In the obese-MASH model, *Pcolce* KO mice exhibited a reduction in the expression of PCPE-1 in the liver (Figs. 3D and EV3E), which was also shown to be associated with diminished liver fibrosis in the HFD mice (Figs. 3E,F and EV3F). In contrast to the BAT *Pcolce* KO mice, liver TG levels were similar between the genotypes in the systemic KO model with dietary obesity (Fig. 3G), with the plasma ALT (Fig. 3H), transcripts for fibrillogenesis, and inflammation shown to be comparable to their littermate controls on respective diets (except for *Tgfb1*, which showed a slight increase in the KO mice with dietary obesity) (Fig. 3H,I). HE staining of BAT, CT scan analysis of abdominal visceral or subcutaneous fat volumes, as well as glucose tolerance

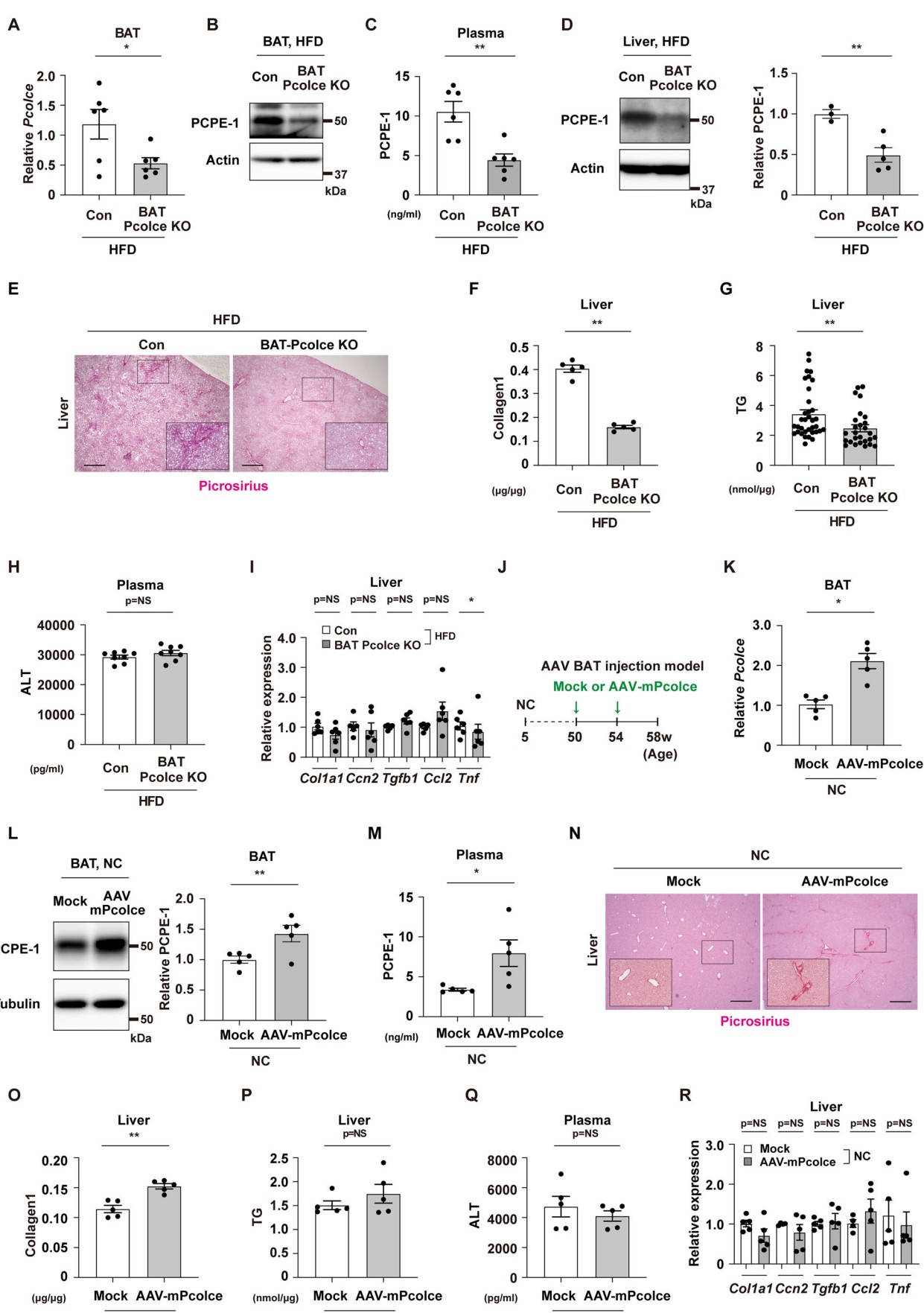

**Figure 2. BATokine PCPE-1 promotes liver fibrosis in MASH.**

Littermate control (Con) or BAT-specific *Pcolce* knockout mice (BAT *Pcolce* KO) were fed a high-fat diet (HFD) from 4 weeks of age and studied at 42–43 weeks of age. (A) Quantitative PCR (qPCR) results showing transcript *Pcolce* in BAT from indicated mice ($n = 6, 6$). (B) Western blot analysis for procollagen C-endopeptidase enhancer-1 (PCPE-1) in BAT from indicated mice. The quantification results for this panel see Fig. EV2C. (C) ELISA for plasma PCPE-1 ($n = 6, 6$) in the indicated mice aged 42–43 weeks. (D) Western blot analysis for PCPE-1 in liver from indicated mice. The right panel indicates the quantification ($n = 3,5$). (E) Picrosirius red staining of liver from the indicated mice. Scale bar $= 500$ μm. For the quantification results see Fig. EV2G. (F–H) ELISA for liver collagen type I (F) ($n = 5, 5$), liver triglyceride (TG) (G) ($n = 35, 27$), and plasma alanine transaminase (ALT) (H) ($n = 8, 8$) in the indicated mice aged 42–43 weeks. (I) Results from qPCR showing transcripts *Col1a1* ($n = 6, 6$), *Ccn2*(Ctgf) ($n = 5, 6$), *Tgfb1* ($n = 5, 6$), *Ccl2* ($n = 5, 6$), and *Tnf* ($n = 6, 6$) in liver from indicated mice. (J) Experimental design of the BAT-adeno-associated virus (AAV) mouse: *Pcolce* gain-of-function model (AAV-mPcolce). AAV-mPcolce was injected into BAT at 50 and 54 weeks of age (a total of 2 injections) in a normal chow (NC) fed mice and studied at 58 weeks of age. (K) Results of a qPCR showing transcript *Pcolce* in BAT ($n = 5, 5$). (L, M) Western blot analysis for PCPE-1 in BAT ($n = 5, 5$) (right panel indicates the quantification of PCPE-1 relative to Tubulin) (L), and ELISA for mouse PCPE-1 in plasma (M) ($n = 5, 5$) of indicated mice. (N) Picrosirius red staining of liver from the indicated mice. Scale bar $= 500$ μm. For the quantification results see Fig. EV2J. (O–Q) ELISA for liver collagen type I (O) ($n = 5, 5$), liver TG (P) ($n = 5, 5$), and plasma ALT (Q) ($n = 5, 5$) from indicated mice. (R) qPCR of *Col1a1* ($n = 5, 5$), *Ccn2*(Ctgf) ($n = 4, 5$), *Tgfb1* ($n = 5, 5$), *Ccl2* ($n = 4, 5$) and *Tnf* ($n = 5, 5$) in liver from the indicated mice. All data were analyzed by an independent-samples T-test. Data information: Representative data of two or more independent series (A, D, F–H), one independent series (I, K, L, R), one independent experiment analyzing samples from at least 2 independently prepared samples (C, F–H, M, O–Q). *$p < 0.05$, **$p < 0.01$. Values represent the mean ± SEM. NS = not significant. All data are from different biological replicates. Source data are available online for this figure.

and insulin tests, showed similar results between the genotypes compared (Fig. EV3G–J; Appendix Figs. S4A and S4B).

Next, a peptide vaccine therapy targeting PCPE-1 (Fig. 3J) was attempted using a protocol that relies on endogenous immune system to produce neutralizing antibodies for the target molecules (Nakagami et al, 2013; Pang et al, 2014; Yoshida et al, 2019). Of the two candidate peptides evaluated, keyhole limpet hemocyanin (KLH)-conjugated KCPSQPRTAA (PCPE-1 vaccine) injection led to a significantly increased PCPE-1 antibody titer (Fig. EV3K). Indeed, this peptide vaccine was shown to reduce plasma levels of PCPE-1 (Fig. 3K) and produce a significant reduction in liver fibrosis (Figs. 3L,M and EV3L). The levels of liver TG (Fig. 3N), plasma ALT (Fig. 3O), transcripts for fibrillogenesis and inflammation (Fig. 3P), body or BAT weight (Fig. EV3M,N), and pathological findings on BAT (Fig. EV3O; Appendix Fig. S4C) showed no significant changes in mice treated with the PCPE-1 vaccine compared to those in controls. These combined results for BAT and systemic *Pcolce* KO mice indicated that PCPE-1 inhibition may contribute to suppression of the fibrotic processes in the livers of mice with dietary obesity.

## ER stress pathway increases PCPE-1 in brown adipocyte

Next, a bioinformatic approach was taken to elucidate the mechanisms involved in the upregulation of PCPE-1 production in BAT (Fig. EV4A). In our obese-MASH model, the levels of phospho-c-Fos and phospho-c-Jun were shown to be increased in BAT (Fig. 4A). Delivery into BAT of two AAVs encoding c-Fos or c-Jun (Fig. 4B) led to an increase in PCPE-1 in both its transcript (Fig. 4C) and protein (Fig. 4D) levels. The overexpression of c-Fos/ c-Jun also led to an increase in PCPE-1 in differentiated brown adipocytes with the AAVs (Fig. 4E,F). Given that ER stress was previously shown to increase c-Fos/c-Jun (Huang et al, 2015), experiments were also conducted to assess for the development of ER stress in BAT in mice with dietary obesity. These demonstrated that ER stress markers including phospho-IRE1 (Serine/threonine-protein kinase/endoribonuclease IRE1) and phospho-JNK (Mitogen-activated protein kinase [c-Jun N-terminal kinase]) increased in BAT in mice with obesity (Fig. 4G,H). It was also shown that phospho-JNK increased with the administration of palmitic acid in differentiated brown adipocytes (Fig. EV4B). Furthermore, the

administration of the ER stress inducer brefeldin A was found to increase phospho-c-Fos and phospho-c-Jun in differentiated brown adipocytes, with these effects abolished with a JNK inhibitor (Fig. 4I,J). It was also found that administration of palmitic acid increased phospho-c-Fos and phospho-c-Jun in differentiated brown adipocytes, again, with these effects abolished with an IRE1 inhibitor (Fig. EV4C). Consistent with previous reports showing that JNK is located downstream of IRE1 (Takeda et al, 2019) and phosphorylates c-Fos and c-Jun (Hammouda et al, 2020; Sundqvist et al, 2019), it was found that the JNK inhibitor reduced the level of phospho-c-Fos and phospho-c-Jun in differentiated brown adipocytes (Fig. EV4D,E). It was also found that palmitic acid and the ER stress inducers, including brefeldin A, tunicamycin, and thapsigargin, increased the transcript *Pcolce* expression in brown adipocytes (Fig. EV4F–I), while treatment with tauroursodeoxycholic acid (TUDCA), an ER stress inhibitor, suppressed the ER stress-induced increase in *Pcolce* in these cells (Fig. 4K). It was also found that both IRE1 and JNK inhibitors reduced *Pcolce* levels in brown adipocytes under ER stress (Fig. 4L). Brefeldin A increased PCPE-1 in the conditioned medium of brown adipocytes, but this secretion was reduced with either inhibitor (Fig. 4M). The brefeldin A-induced increase in *Pcolce* was also reduced with an AP1 inhibitor in these cells (Fig. 4N). Finally, it was found that the in vivo administration of this TUDCA reduced levels of the transcript *Pcolce* in BAT (Fig. 4O) and the plasma levels of PCPE-1 (Fig. 4P). Taken together, our results show that high-calorie diet-induced ER stress increases PCPE-1 production in BAT through the activation of the IRE-1/JNK/c-Fos/c-Jun pathway and enhances liver fibrosis (Fig. 4Q), suggesting that inhibition of the BATokine PCPE-1 may represent a therapeutic modality for MASH.

## Discussion

In this study, we have shown that the BATokine PCPE-1 enhances liver fibrosis in mice with dietary obesity. Combined results from quantitative PCR studies and the BAT-specific *Pcolce* KO mice provide evidence that PCPE-1 is an adipokine produced predominantly from BAT in mice. In our obese-MASH model, BAT-specific *Pcolce* depletion resulted in the inhibition of fibrosis in the liver. Two additional models involving PCPE-1 suppression, i.e., a

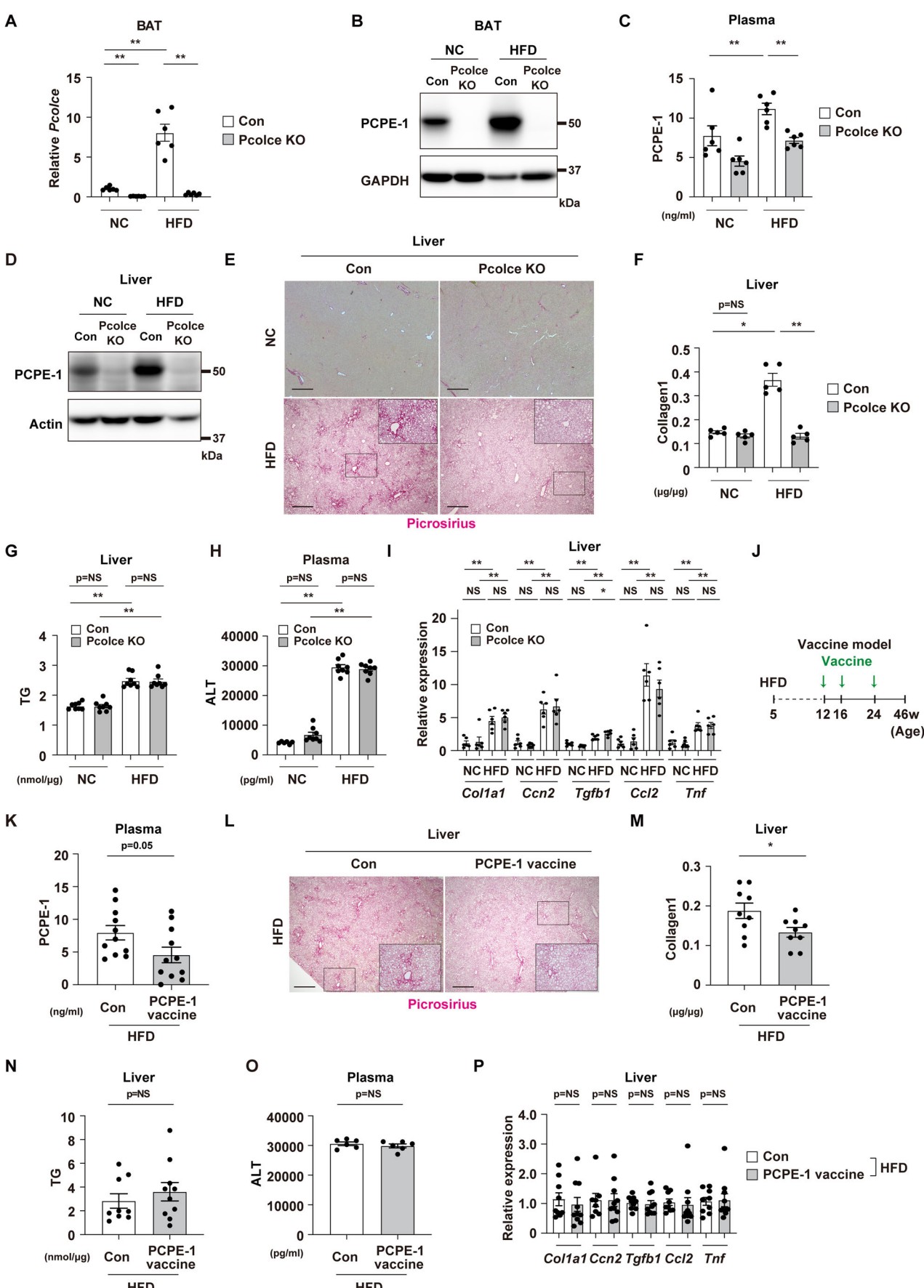

**Figure 3.   Models with systemic PCPE-1 inhibition ameliorate liver fibrosis.**

Littermate control (Con) or systemic *Pcolce* KO mice were fed a normal chow (NC) or a high-fat diet (HFD) from 4 weeks of age and studied at 38–40 weeks of age. **(A)** Quantitative PCR (qPCR) results showing transcript *Pcolce* in BAT from indicated mice ($n = 6, 6, 6, 6$). **(B–D)** Western blot analysis for PCPE-1 in BAT **(B)** or liver **(D)**, and ELISA showing plasma PCPE-1 **(C)** ($n = 6, 6, 6, 6$) from indicated mice. For the quantification of these Western blot results see Fig. EV3B or Fig. EV3E. **(E)** Picrosirius red staining of liver from the indicated mice. Scale bar = 500 μm. For the quantification results see Fig. EV3F. **(F–H)** ELISA for liver collagen type I **(F)** ($n = 5$, 5, 5, 5), liver triglyceride (TG) **(G)** ($n = 8, 8, 8, 8$), and plasma alanine transaminase (ALT) **(H)** ($n = 6, 8, 8, 8$) **(G)** in the indicated mice. **(I)** Results from qPCR showing transcript *Col1a1*, *Ccn2*(Ctgf), *Tgfb1*, *Ccl2*, and *Tnf* in liver from indicated mice ($n = 6, 6, 6, 6$). **(J)** Experimental design of a vaccination therapy targeting PCPE-1. C57BL/6NCrSlc mice were fed a HFD from 4 weeks of age. These mice were injected with a PCPE-1 vaccine at 12, 16, and 24 weeks of age (total of 3 injections), and studied at 46 weeks of age. **(K)** ELISA for plasma PCPE-1 in the indicated mice ($n = 11, 11$). **(L)** Picrosirius red staining of liver from the indicated mice. Scale bar = 500 μm. For the quantification results see Figure EV3L. **(M–O)** ELISA for liver collagen type I **(M)** ($n = 9, 9$), liver TG **(N)** ($n = 9, 10$), and plasma ALT **(O)** ($n = 6, 6$) in the indicated mice. **(P)** Results from qPCR showing transcripts *Col1a1* ($n = 9, 10$), *Ccn2*(Ctgf) ($n = 8, 10$), *Tgfb1* ($n = 9, 10$), *Ccl2* ($n = 9, 10$) and *Tnf* ($n = 9, 10$) in liver from indicated mice. All data were analyzed by an independent-samples T-test. Data information: Representative data of two or more independent series **(A, I)**, one independent series **(P)**, one independent experiment analyzing samples from at least 2 independently prepared samples **(C, F–H, K, M–O)**. *$p < 0.05$, **$p < 0.01$. Values represent the mean ± SEM. NS = not significant. All data are from different biological replicates. Source data are available online for this figure.

systemic *Pcolce* KO model and a PCPE-1 peptide-vaccinated model, both showed diminished fibrosis in the liver. In contrast, experimentally induced expression of PCPE-1 in BAT led to an increase in liver fibrosis, indicating the therapeutic potential of this secreted protein in MASH. Of the adipocytes evaluated in this study, BAT, which was initially characterized to have roles in thermogenesis, is now shown to be involved in the regulation of systemic metabolism, in a manner that is similar to WAT via the production of WAT adipokines that mediate systemic effects in remote organs. In 1993, TNF-α, an adipokine produced from WAT was identified by Hotamisligil et al as the likely cause of systemic insulin resistance in obese rats (Hotamisligil et al, 1993) and this was followed by the discovery of IL-1β and IL-6 as other WAT-derived adipokines shown to have roles in promoting inflammation in systemic organs (Ouchi et al, 2011). Thus, the rationale for targeting these adipokines in obesity-linked diseases. Indeed, Ridker et al conducted a randomized double-blind trial enrolling 10,061 patients with a history of myocardial infarction and a high sensitivity C-reactive protein level of 2 mg or more per liter and showed that canakinumab, a neutralizing antibody for IL-1β, led to a significant lower rate of recurrent cardiovascular events in these patients (Ridker et al, 2017). The ZEUS study is now underway to evaluate the effect of IL-6 inhibition with ziltivekimab in patients with cardiovascular disease, chronic kidney disease and inflammation (ClinicalTrials.gov Identifier: NCT05021835). Thus, adipokine studies in WAT appear to represent an approach aimed at translating basic research into clinical applications. Indeed, recent studies have identified several BAT-derived adipokines as contributors to maintenance of homeostasis in vessels or pancreatic beta cells (Cataldo et al, 2022; Shi et al, 2022). Thus, these findings have opened new avenues for BAT research focusing on the endocrinal aspect of this organ. However, our understanding of BAT-derived adipokines, i.e., BATokines, is still limited, compared to that of WAT-derived adipokines.

In this study focused on secreted molecules, it was found that the transcript *Pcolce*, the coding gene for PCPE-1, was increased in BAT in mice with obesity. Given that it binds to the C-terminal propeptide of type I procollagen and enhances the activity of procollagen C-proteinase, PCPE-1 has a critical role in the production of mature and structured collagen fibrils and is a key regulator of tissue fibrosis. Studies in rodents and humans showed that circulating PCPE-1 is associated with liver fibrosis (Hassoun et al, 2017; Hassoun et al, 2016) and systemic depletion of PCPE-1

ameliorates fibrosis in the liver in a murine MASH model (Sansilvestri Morel et al, 2022), indicating that PCPE-1 may represent a relevant biomarker as well as a therapeutic target for MASH (Lagoutte et al, 2021). In the current study, it was found that BAT-specific PCPE-1 depletion resulted in a significant reduction of this molecule in circulation and ameliorated fibrosis in the liver in mice with dietary obesity. In contrast, BAT-specific PCPE-1 induction increased circulating PCPE-1 and enhanced liver fibrosis in mice fed a normal chow diet and a HFD. In two additional models involving systemic PCPE-1 suppression, i.e., a systemic *Pcolce* KO model and PCPE-1-targeted peptide vaccine model, generated to test a therapeutic proof of concept, it was shown that both models were associated with diminished liver fibrosis.

Multiple mechanisms are involved in the pathogenesis of MASH and include excessive lipid influx and chronic sterile inflammation in the liver. Again, while MASH patients with advanced fibrosis are at high risk for morbidity and mortality from the disease; however, at this stage, there are no robust data showing the benefit of antifibrotic therapy in the clinical outcome of patients with MASH (Harrison et al, 2023). Thus, the combination therapy approach may be critical for the treatment of MASH, including modulators of the energy imbalance, inflammation, and fibrosis. Our three models with PCPE-1 loss of function, as well as our model with PCPE-1 gain of function, showed no change in the level of ALT in the plasma. Transcript studies also showed comparable results for markers for inflammation in the groups compared, except for *Tnf*, which was shown to be reduced in the BAT *Pcolce* KO mice. Interestingly, liver TG levels were shown to be significantly reduced in our BAT-*Pcolce* KO model with dietary obesity, while *Pcolce* overexpression in BAT tended to increase liver TG levels in our AAV-mPcolce-induced gain-of-function model and significantly increased liver TG levels under obese conditions. To further test PCPE-1 for its potential role in lipid metabolism, primary hepatocytes were treated, under steatotic conditions with a recombinant PCPE-1 protein. PCPE-1 treatment did not alter the expression profiles of the genes tested, except for *Ldlr*, which was shown to be reduced in the steatosis group. Again, intracellular TG levels were increased in steatotic hepatocytes, while this change was not affected by the PCPE-1 treatment. Our data indicate the potential pleiotropic effect of PCPE-1 and/or fibrosis in lipid metabolism; however, further studies are needed to test this hypothesis. While the mechanisms responsible for the progression of liver fibrosis remain unclear, our data indicate a crucial role for

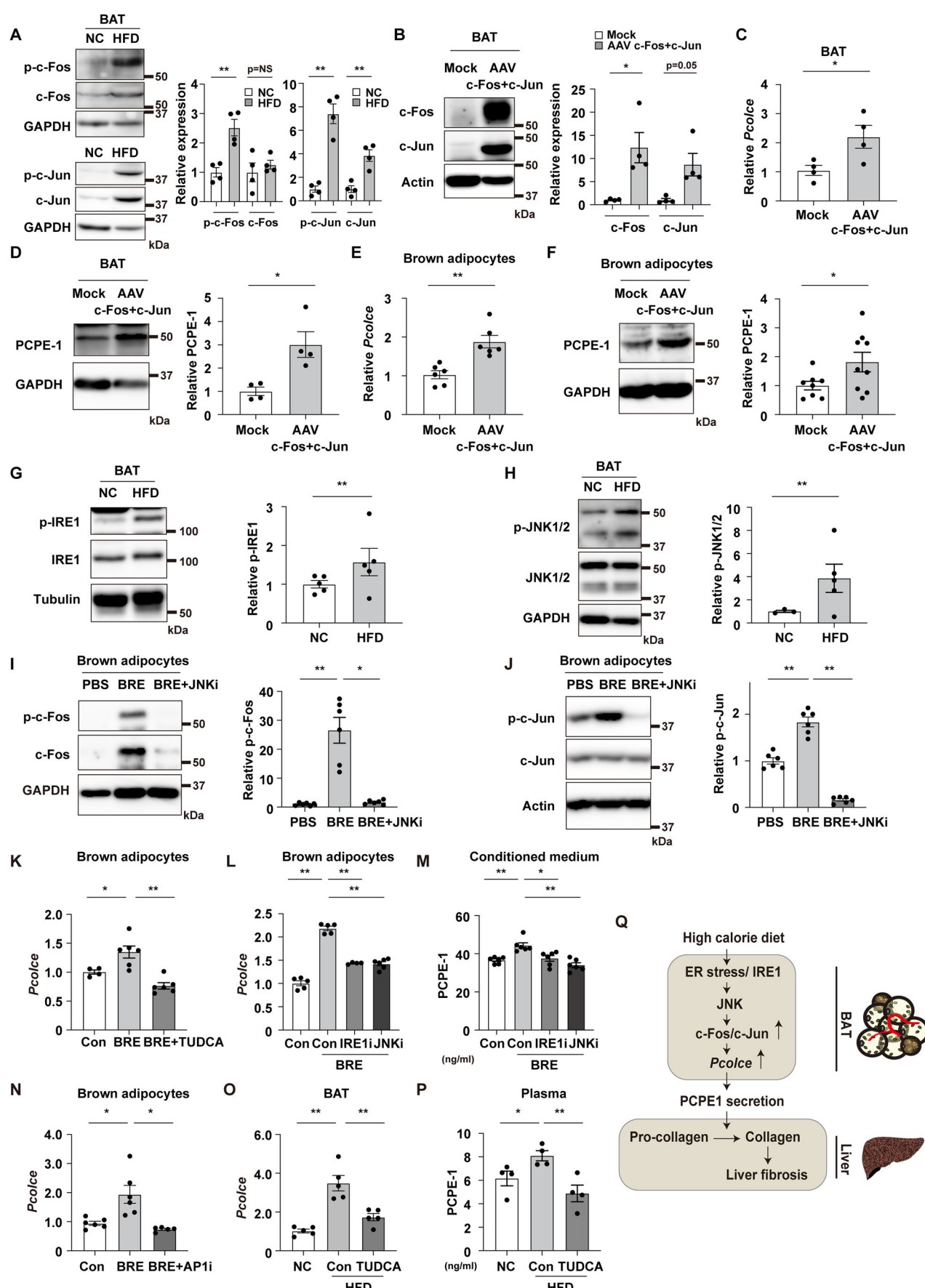

**Figure 4. ER stress pathway increases PCPE-1 in brown adipocytes.**

(A) Western blot analysis for phospho c-Fos (p-c-Fos), c-Fos, phospho c-Jun (p-c-Jun), and c-Jun in brown adipose tissue (BAT) from C57BL/6NCrSlc wild-type (WT) mice fed a normal chow (NC) or a high-fat diet (HFD) (aged 32–38 weeks). The right panel indicates the quantification of p-c-Fos, c-Fos, p-c-Jun, and c-Jun relative to GAPDH ($n = 4, 4$). (B) Western blot analysis for c-Fos and c-Jun in BAT injected with adeno-associated virus (AAV) encoding c-Fos (AAV-c-Fos) co-injected with AAV-c-Jun (AAV-c-Fos+c-Jun) into this tissue. AAV study was done in 12-week-old WT mice maintained on a normal chow diet 7 days after the injection. The right panel indicates the quantification of c-Fos and c-Jun relative to Actin ($n = 4, 4$). (C, D) Results from quantitative PCR (qPCR) showing transcript *Pcolce* ($n = 4, 4$) (C) and western blot analysis for PCPE-1 (D) in BAT from indicated mice. The right panel in (D) indicates the quantification of PCPE-1 relative to GAPDH ($n = 4, 4$). (E, F) Results from qPCR showing transcript *Pcolce* ($n = 6, 6$) (E) and western blot analysis for PCPE-1 (F) in differentiated brown adipocytes introduced with mock or AAV-c-Fos+c-Jun. In Fig. 4F, the right panel indicates the quantification of PCPE-1 relative to GAPDH ($n = 8, 9$). (G, H) Western blot analysis for p-IRE1, IRE1 (G), and p-JNK1/2 and JNK1/2 (H) in BAT from WT mice fed a NC or a HFD (34–38 weeks of age). The right panel indicates the quantification of p-IRE1 relative to Tubulin (G) ($n = 5, 5$) and p-JNK1/2 (H) ($n = 3, 5$) relative to GAPDH in BAT. (I, J) Western blot analysis for p-c-Fos, c-Fos (I), p-c-Jun, and c-Jun (J) in fully differentiated brown adipocytes administrated with PBS, BRE (Brefeldin A, as an ER stress inducer), or BRE + JNK inhibitor (SP600125). The right panels in (I) and (J) indicate the quantification of p-c-Fos relative to GAPDH ($n = 6, 6, 6$) or p-c-Jun relative to Actin ($n = 6, 6, 6$). (K, L) Results from qPCR showing transcript *Pcolce* in differentiated brown adipocytes. These cells were administrated with BRE, together with an introduction of tauroursodeoxycholic acid sodium salt (TUDCA) (K) ($n = 4, 6, 6$), IRE1 inhibitor (IRE1-i) or JNK inhibitor (JNK-i) (L) ($n = 5, 5, 4, 6$). (M) ELISA study testing PCPE-1 in conditioned medium of differentiated brown adipocytes treated with the indicated groups ($n = 6, 6, 6, 6$). (N) Results from qPCR showing transcript *Pcolce* in differentiated brown adipocytes. These cells were administrated with BRE, together with an introduction of AP1 inhibitor (AP1-i) (N) ($n = 6, 6, 5$). (O) Results from qPCR showing transcript *Pcolce* in BAT of the indicated mice ($n = 5, 5, 5$). (P) ELISA study testing plasma PCPE-1 in the indicated group ($n = 4, 4, 4$). (Q) Scheme summarizing ER stress/IRE1/JNK/c-Fos/c-Jun signaling up-regulates PCPE-1 expression in BAT under high-calorie diet. c-Fos/c-Jun increases the production of PCPE-1 in BAT and promotes fibrosis in the liver. All data, except for figures (I–O) were analyzed by an independent-samples T-test. The data in (I–O) were analyzed by a two-way analysis of variance (ANOVA) followed by Tukey's multiple comparison test (K, M, O), Dunnett test (J, L), or non-parametric Kruskal Wallis test (I, N). Data information: Representative data of two or more independent series (A, B, D, E–L, N), one independent series (C, M, O, P). *$p < 0.05$, **$p < 0.01$. Values represent the mean ± SEM. NS = not significant. All data are from different biological replicates. Source data are available online for this figure.

BAT-derived PCPE-1 in the progression of the fibrotic process in this organ. MASH poses challenges, in that it leads to liver cirrhosis and liver cancer, while liver fibrosis represents a risk factor for liver cancer. Thus, a question that remains to be addressed is whether suppression of fibrosis per se might inhibit the transition to liver cancer. Further studies are needed to test this hypothesis.

Mechanistically, it was found that high-calorie, diet-induced ER stress increases PCPE-1 production in BAT through the activation of the IRE-1/JNK/c-Fos/c-Jun pathway. In our dietary obese mice, ER stress markers, including phospho-IRE-1 and phospho-JNK1/2, were shown to be increased in BAT. With regard to mitochondrial uncoupling protein 1 (UCP-1), which was previously shown to have a central role in thermogenic response in BAT and whose level was reported to decline in this organ with obesity (Shimizu et al, 2014), it was also shown that its deficiency promoted ER stress in BAT (Bond et al, 2018). Both these previous reports and our current findings suggest that ER stress develops in BAT with excessive caloric intake. The phosphorylation c-Fos and c-Jun, located downstream of ER stress signaling, was increased in BAT in mice with obesity. c-Fos dimerizes with c-Jun to form a complex called activator protein 1 (AP-1), and a previous report demonstrated that c-Fos in the central nervous system is involved in the sympathetic activation of BAT (Zhang et al, 2011). However, the role of this protein in BAT was previously unknown. In our study, it was shown that suppression of ER stress led to a reduction in the transcript *Pcolce* in BAT, as well as in the circulatory level of this protein in a model of diet-induced obesity. In contrast, ER stress led to an increase in the transcript *Pcolce* in brown adipocytes, while this increase was reduced with an ER stress inhibitor or an AP-1 inhibitor, suggesting that suppression of this signaling pathway in BAT may represent one viable approaches to reducing circulating PCPE-1. However, challenges remain in the development of tissue-specific therapies targeting this pathway, and potential off-target systemic effects should be an issue. Previous studies in rodents have shown that molecules, including angiotensin II (AngII), dipeptidyl peptidase-4 (DPP4), and

semaphorin3E, are suppressed with vaccines targeting these molecules, thus ameliorating the pathogenesis of hypertension or diabetes (Nakagami et al, 2013; Pang et al, 2014; Yoshida et al, 2019). Our results suggested that a vaccine approach could likely achieve therapeutic PCPE-1 inhibition. Analysis of transcriptome and transgenic mice data in this study showed that PCPE-1 was mainly produced from BAT in mice. Our quantitative PCR study and analysis of GSE123394 showed that PCPE-1 is also produced from the liver and increased with dietary obesity. While at this stage we were unable to generate and test a liver-specific PCPE-1 depletion model, which is a limitation of our study, transcriptome studies from in vivo and in vitro samples indicate that the contribution of liver-derived PCPE-1 may be rather small. While the source of this molecule in humans remains unclear, this may not raise major issues for drug development, given that PCPE-1 is a secreted protein. Patients with MASH are shown to exhibit increased circulatory levels of PCPE-1, suggesting that the suppression of this pro-fibrotic protein may represent a novel treatment modality for MASH. Finally, a fundamental question that remains to be answered is why this molecule is predominantly produced from BAT. Thus, further studies are required to comprehensively understand the role of this molecule under both physiological and pathological conditions.

## Methods

### Human samples

Blood samples and clinical data were collected from patients of Niigata University Hospital. All subjects provided written informed consent prior to participation in these studies. The Scientific-Ethics Committees and Institutional Review Board of National Cerebral and Cardiovascular Center, Niigata University and Juntendo University approved the protocols of all the studies (protocol number R23028, G2018-0023, 2015-2193, E21-0075-H01, E22-0003-M01), and each investigation was

performed in accordance with the Declaration of Helsinki. Serum samples were subjected to ELISA, and PCPE1 ELISA kit:: Human Procollagen C Proteinase Enhancer 1 ELISA Kit, 96-Strip-Wells-(Sandwich) was used under manufacturer's instruction (MyBioSource, MBS722836S).

## Animal models

All animal experiments were conducted in compliance with the protocol, which was reviewed by the Institutional Animal Care and Use Committee of Niigata University, Juntendo University, and National Cerebral and Cardiovascular Center Research Institute and approved by the presidents of those universities. Mice were housed in the animal facilities under specific pathogen-free conditions at a constant temperature of 23 °C, and a 12 h light/12 h dark cycle. All experiments analyzed male mice. C57BL/6NCrSlc male mice were purchased from SLC Japan (Shizuoka, Japan). Some of these mice were maintained on a high-fat diet (HFD) (HFD32, CLEA Japan) or normal chow from 4 to 34–40 weeks of age. Systemic *Pcolce* knockout or floxed *Pcolce* mice were generated by Abe et al, To generate *Pcolce*-flox mice, a targeting vector was constructed as follows: a 1.11 kb DNA fragment carrying exon 3 and exon 4 of the *Pcolce* was amplified by PCR, and inserted into the KpnI sites of the middle entry clone (pDME-1). In this clone, a DNA fragment of pgk promoter-driven Neo-poly(A) (pgk-Neo) flanked by two frt sites and loxP sequence was located at the site 278 bp downstream of the exon 4, while the other loxP sequence was placed at the site 345 bp upstream of the exon 3. The 6.97 kb upstream and 5.52 kb downstream homologous genomic DNA fragments were retrieved from the BAC clone, and then subcloned to 5′ entry clone (pD5UE-2) and 3′ entry clone (pD3DE-2), respectively. For targeting vector assembly, the three entry clones were recombined to a destination vector plasmid (pDEST-DT; containing a cytomegalovirus enhancer/chicken actin (CAG) promoter-driven diphtheria toxin gene) by using MultiSite Gateway Three-fragment Vector construction Kit (Thermo Fisher Scientific). Culture of ES cells and generation of chimeric mice were carried out as previously described (Mishina and Sakimura, 2007).

To generate *Pcolce*-deficient mice by CRISPR-Cas technology, hU6-sgRNA plasmids (Yuza et al, 2018) were ligated synthetic oligonucleotides of the sequence 5′-CTCCAGAGCATCG-TAGCGGC-3′ and 5′-GGCACCTCAGGCCAGCGACT-3′, which are identical to part of exon3 of Pcolce. C57BL6/N female mice were superovulated and mated with C57BL6/N males, and fertilized eggs were collected from the oviduct. The pronuclear stage eggs were injected with a plasmid expressing Cas9 nickase (System Bioscience) and hU6-sgRNA plasmids. Ucp1-Cre$^{+/-}$ (C57BL/6J background) were crossed with mice carrying floxed Pcolce alleles with a C57BL/6 background to generate mice with brown adipose tissue (BAT) specific PCPE-1 knockout (Ucp1-Cre$^{+/-}$; *Pcolce*$^{fl/fl}$) (BAT *Pcolce* KO). The genotypes of the littermate controls were Ucp1-Cre$^{-/-}$: *Pcolce*$^{fl/fl}$. For study with the peptide vaccination, KLH-conjugated antigenic peptide (20 μg) and an equal volume of complete or incomplete Freund's adjuvant (Sigma-Aldrich: F5881 (complete) or F5506 (incomplete)) were emulsified by vortexing. For immunization, HFD-feeding mice were injected subcutaneously with the KLH-conjugated peptide at the age of 12 weeks (with complete Freund's adjuvant), 16 weeks (with incomplete Freund's adjuvant), and 24 weeks (with incomplete Freund's

adjuvant). Control groups received an equal volume of KLH mixed with complete or incomplete Freund's adjuvant. Plasma was collected from the tail vein of each mouse for analysis of antibody titers (Yoshida et al, 2019). To generate BAT delivered mouse *Pcolce* gain-of-function model, pAAV-mouse *Pcolce* was directly injected into BAT of a NC or a HFD fed male mice at the age of 50 and 54 weeks (totally 2 injections) with a concentration of $5 \times 10^{10}$ GC/mice. The mock controls were injected with AAV-EGFR at the same concentration. Physiological studies were started 30 days after the second injection. To generate BAT c-Fos and c-Jun gain-of-function model, pAAV-mouse c-fos, and pAAV-mouse c-jun were directly injected into BAT at a concentration of $5 \times 10^8$ GC/mice in NC male mice (12 weeks of age). Physiological studies were started 7 days after injection (Shimizu et al, 2014). C57BL/6NCrSlc male mice maintained on a HFD were intraperitoneally injected with TUDCA (100 mg/kg) every 2 days for 1 week (total 3 injections) and subjected to sample collection at 29 weeks of age. Investigators were blinded to mouse genotypes during experiments. The study was carried out in compliance with the ARRIVE guidelines.

## Physiological analysis

CT scan (LaTheta LCT-200, Hitachi, Japan) at 2-mm slice intervals was performed for whole body, and volume of visceral fat or subcutaneous fat were analyzed according to the manufacturer's protocol. For the intraperitoneal glucose tolerance test (IGTT), mice were starved for 4 h and were given glucose intraperitoneally at a dose of 1 g/kg (body weight) (D-(+)-Glucose solution (Sigma: G8644)) in the early afternoon. For the insulin tolerance test (ITT), mice were given human insulin intraperitoneally at a dose of 1U/kg (body weight)(Humulin R (100 Unit/ml): Lilly) at 1:00 p.m. without starvation. Tail vein blood was collected at 0, 15, 30, 60, and 120 min after administration, and blood glucose levels were measured with a glucose analyzer (Sanwa Kagaku Kenkyusho CO., LTD).

## Cell culture and in vitro studies

The mouse brown pre-adipocyte cell line was a kind gift from Dr. C. Ronald Kahn (Joslin Diabetes Center and Harvard Medical School, Section on Integrative Physiology and Metabolism, Boston, USA) (Klein et al, 1999). The cell line was established from wild-type FVB mice, and was immortalized by infection with the retroviral vector pBabe encoding SV40T antigen. Cells were cultured in high glucose DMEM with 10% FBS and 100 U/ml P/S, and differentiation was induced as described previously (Fasshauer et al, 2001). Fully differentiated brown adipocytes were used for further analysis after 10 days of culture. The following drugs were used for fully differentiated brown adipocytes in vitro experiments: Palmitic acid (100 μM for 1–12 h, depending on the assay; Sigma-Aldrich, #P0500), SP600125 (2 μM: Sigma-Aldrich, #S5567) as a JNK inhibitor, Brefeldin A (BRE: 2.5 μM for 6 h: Sigma-Aldrich, #B6542), Tunicamycin (1 μg/ml for 6 hr: Sigma-Aldrich, #T7765), Thapsigargin (100 nM for 6 h; Sigma-Aldrich, #T9033) as ER stress inducers, STF-083010 (30 μM: Sigma-Aldrich, #SML0409) as an IRE1 inhibitor, T-5224 (10 μM: MedChemExpress (MCE), #HY-12270) as an AP1 inhibitor, Tauroursodeoxycholic acid dihydrate (TUDCA) (100 μM: Tokyo chemical industry CO, #T1567) as an ER stress inhibitor. The JNK inhibitor, IRE1 inhibitor, AP1 inhibitor, and ER stress inhibitor were

administrated 1 h before palmitic acid or BRE treatment in in vitro studies. To generate c-Fos and c-Jun overexpression brown adipocytes, the fully differentiated brown adipocytes were infected with AAV-mouse c-fos (200 GC/cell) and AAV-mouse c-jun (200 GC/cell) for 6 h (qPCR analysis) and 12 h (Western blot analysis). Primary cardiac fibroblasts and brown adipocytes were isolated from 2-day-old rat neonatal (Wistar rat) as previously reported (Galmozzi et al, 2021; Kumar et al, 2023). Primary hepatocytes and brown adipocytes were isolated from adult C57BL/6NCrSlc male mice (10–12 weeks of age) as previously reported (Charni-Natan and Goldstein, 2020; Galmozzi et al, 2021). Mouse hepatocyte cell line: AML-12 (ATCC, #CRL-2254) was cultured according to the manufacturer's instructions. Steatotic AML-12 was generated by maintaining cells with 4% BSA + 200 μM Oleic acid (Wako, #155-03401) and 200 μM Linoleic acid (Wako, #126-03612) as previously reported (Nativ et al, 2013), and recombinant mouse PCPE-1 protein (R&D System, #2239-PE, 400 ng/ml, 6 days) was added into these cells.

## Design and synthesis of the vaccine

To generate neutralizing antibodies for mouse PCPE-1, two antigenic peptide sequences (spanning amino acids 459–468 (KCPSQPRTAA) and amino acids 291–300 (RGPQSRSDPK) of mouse PCPE-1) were selected on the basis of previous reports (Pang et al, 2014). The N-terminus or lysine of each candidate peptide was conjugated to keyhole limpet hemocyanin (KLH) via glutaraldehyde, and the synthetic peptides were purified by reverse-phase high-performance liquid chromatography (>99% purity) (Peptide Institute Inc., Osaka, Japan), as described previously (Nakagami et al, 2013).

## Antibody titer measurement

The antibody titer generated by injection of each peptide was measured by ELISA. A 96-well microtiter plate (Nunc-Immuno MicroWell 96-well solid plate, Thermo Scientific) was coated with one of the antigenic peptides conjugated to bovine serum albumin (BSA) in carbonate buffer. After coating, plasma collected from pre- and post-immunized mice was added to the wells and incubated overnight. Then the plate was incubated with the secondary antibody (ECL Anti-mouse IgG, Horseradish Peroxidase linked whole sheep antibody, GE Healthcare). After reaction with the TMB substrate (Sigma-Aldrich, T0440), the optical density was measured at 450 nm (OD450) by using a microtiter plate reader (iMark microplate reader, Bio-Rad).

## Adeno-associated virus (AAV)

The purified pAAV-mouse Pcolce or pAAV-EGFR (negative control) recombinant AAVDJ virus were made from pAAV[Exp]-CMV>mPcolce[NM_008788.2]:WPRE vector (VectorBuilder, AAVdjLP(VB211222-1285kyq)-C) or pAAV[Exp]-CMV > EGFR:WPRE vector (Vector-Builder, AAVdjLP(VB010000-9394npt)-C). The pAAV-mouse c-fos, pAAV-mouse c-jun, and pAAV-negative control vectors were constructed using standard subcloning techniques according to the manufacturer's instructions (VPK-413-DJ and AAV-DJ Helper Free Expression System, Cell Biolabs Inc, VPK-410-DJ). We co-transfected HEK293 cells with pAAV expression vector, pAAV-DJ, and pHelper using a transfection reagent (X-tremeGENE9 DNA Transfection

Reagent, Roche, 06365809001). AAV was harvested from these cells by freeze and thaw cycles, and purified with a ViraBind AAV Purification Kit (Cell Biolabs Inc, VPK-140). The Titer of purified AAV was quantified with QuickTiter AAV Quantitation Kit (Cell Biolabs Inc, VPK-145).

## RNA analysis

Total RNA (1 μg) was isolated from tissue and cell samples with QIAzol Lysis Reagent (QIAGEN). Then, qPCR was performed with QuantStudio6 (Applied Biosystems) using PowerUp SYBR Green Master Mix (Applied Biosystems) according to the manufacturer's instructions. To quantify the copy number, we used Light Cycler 480 software version 1.5 (Roche) and performed a calculation by the Fit Points method according to the manufacturer's instructions.

The following primers were employed, with *Actb* or *Rplp0* as the internal control:

*Actb*; 5′-CTAAGGCCAACCGTGAAAAG-3′, 5′-ACCAGAGG-CATACAGGGACA-3′

*Rplp0*; 5′-GATGCCCAGGGAAGACAG-3′, 5′-ACAATGAAGC ATTTTGGATAA-3′

*Pcolce*; 5′-CTAAATCCCAACCTGCAGAGA-3′, 5′-TGACCGC TTGTACTGCTTTG-3′

*Col1a1*; 5′-CATGTTCAGCTTTGTGGACCT-3′, 5′-GCAGCTG ACTTCAGGGATGT-3′

*Ccn1* (Cyr61); 5′-CACTGAAGAGGCTTCCTGTCTT-3′, 5′-CC AAGACGTGGTCTGAACGA-3′

*Ccn2* (Ctgf); 5′-CAATGCACTTGCCTGGATGG-3′, 5′-TTCC AGTCGGTAGGCAGCTA-3′

*Adamts2*; 5′-AGCCACCATCTGTCTGTCAAG-3′, 5′-GGTGA TGGTTCCGTGGAGTG-3′

*Tgfb1*; 5′-TGGAGCAACATGTGGAACTC-3′, 5′-CAGCAGCC GGTTACCAAG-3′

*Ccl2*; 5′-CATCCACGTGTTGGCTCA-3′, 5′-GATCATCTTGC TGGTGAATGAGT-3′

*Tnf*; 5′-TCTTCTCATTCCTGCTTGTGG-3′, 5′-GGTCTGGG CCATAGAACTGA-3′

*Cd36*; 5′-TTGTACCTATACTGTGGCTAAATGAGA-3′, 5′-CT TGTGTTTTGAACATTTCTGCTT-3′

*Srebf1*; 5′-GGTTTTGAACGACATCGAAGA-3′, 5′-CGGGAA GTCACTGTCTTGGT-3′

*Ldlr*; 5′-CCGGAGTTGCAGCAGAAGA-3′, 5′-TCAGGAATGC ATCGGCTGAC-3′

*Nr1h3*; 5′-TGCGCTCAGCTCTTGTCAC-3′, 5′-CTCCGTTGCA GAATCAGGAGAA-3′

*Gpam*; 5′-GGAAGGTGCTGCTATTCCTG-3′, 5′-TGGGATAC TGGGGGTTGAAAA-3′

*Dgat2*; 5′-GGCGCTACTTCCGAGACTAC-3′, 5′-TGGTCAGC AGGTTGTGTGTC-3′

*Acaca*; 5′-GCGTCGGGGTAGATCCAGTT-3′, 5′-CTCAGTGGG GCTTAGCTCTG-3′

*Fasn*; 5′-GCTGCTGTTGGAAGTCAGC-3′, 5′-AGTGTTCGTT CCTCGGAGTG-3′

*Pnpla2*; 5′-TGACCATCTGCCTTCCAGA-3′, 5′-TGTAGGTG GCGCAAGACA-3′

*Lipe*; 5′-GCGCTGGAGGAGTGTTTTT-3′, 5′-CCGCTCTCCA GTTGAACC-3′

*Pnpla3*; 5′-TGGGAACCTCCAACTTCTGA-3′, 5′-ATAGCACA GCTCTCCCATCAC-3′

*Cebpa*; 5′-CGCTGGTGATCAAACAAGAG-3′, 5′-GGTGGCTG
GTAGGGGAAG-3′

*Pparg*; 5′-AAGACAACGGACAAATCACCA-3′, 5′-GGGGGTG
ATATGTTTGAACTTG-3′

*Cpt1a*; 5′-GACTCCGCTCGCTCATTC-3′, 5′-TCTGCCATCTT
GAGTGGTGA-3′

## Public database studies

Transcriptome studies were analyzed at the Gene Expression Omnibus (https://www.ncbi.nlm.nih.gov/geo/), or Reference Expression Dataset (RefEx) (https://refex.dbcls.jp/index.php?lang=en) platforms. Single-cell RNAseq data deposited in the Tabula Muris Senis (https://tabula-muris-senis.ds.czbiohub.org) was studied at the platform.

## Western blot analysis

Whole-cell or tissue lysates were prepared in lysis buffer (10 mM Tris-HCl, pH 8, 140 mM NaCl, 5 mM EDTA, 0.025% NaN$_3$, 1% Triton X-100, 1% deoxycholate, 0.1% SDS, 1 mM PMSF, 5 μg ml$^{-1}$ leupeptin, 2 μg ml$^{-1}$ aprotinin, 50 mM NaF, and 1 mM Na$_2$VO$_3$). Then the lysates (40–50 μg) were resolved by SDS-PAGE. Proteins were transferred to a PVDF membrane (Millipore) that was incubated with following primary antibodies: anti-PCPE-1 antibody (R&D Systems: MAB2239 (Figs. 2D and 4F), Lifespan Biosciences: LS-C111903 (all other figures)), anti-phospho-c-Fos (Ser32) antibody (Cell Signaling, 5348), anti-c-Fos antibody (Cell Signaling, 2250), anti-phospho-c-Jun (Ser63) antibody (Cell Signaling, 9261), anti-c-Jun antibody (Cell Signaling, 9165), anti-phospho-IRE1 (Ser724) antibody (Affinity Biosciences, AF7150), anti-IRE1 antibody (Abcam, ab37073), anti-phospho-SAPK/JNK (Thr183/Tyr185) antibody (Cell Signaling, 4668), anti-SAPK/JNK antibody (Cell Signaling, 9252), anti-GAPDH antibody (Proteintech, 10494-1-AP), anti-Pan-Actin antibody (Cell Signaling, 4968), anti-α-Tubulin antibody (Cell Signaling, 2125). All primary antibodies were used at a dilution of 1:1000. Subsequently, incubation was done with horseradish peroxidase-conjugated goat anti-rabbit IgG(H + L) (Jackson Immunoresearch, #111-035-003), goat anti-rat IgG(H + L) (#112-035-003), and rabbit anti-goat IgG(H + L) (#305-035-003) at a dilution of 1:5000. After incubation, proteins were detected by enhanced chemiluminescence (GE).

## ELISA

The following kits were used for ELISA according to the instructions of the manufacturer: Mouse Procollagen C Proteinase Enhancer 1 ELISA Kit (My BioSource, MBS021229), Human Procollagen C Proteinase Enhancer 1 ELISA Kit (My BioSource, MBS722836S), Mouse Collagen Type I ELISA Kit (My BioSource, MBS724458), ALT (Alanine Transaminase) ELISA Kit (My BioSource, MBS2500279), and Triglyceride (TG) Assay Kit (Abcam, ab65336).

## Histological analyses and immunostaining

Samples of brown adipose tissue (BAT) and liver were harvested, fixed overnight in 10% formalin, embedded in paraffin, and sectioned for immunofluorescence or hematoxylin-eosin (HE) or picrosirius red staining (Shimizu et al, 2013), and photographed with a Biorevo (Keyence Co., Osaka, Japan). For immunostaining, deparaffinized sections were retrieved with citrate buffer and incubated with anti-mouse PCPE-1 antibody (R&D Systems, MAB2239) at 1:50 dilution. Anti-Rat IgG Cy5-conjugated (Invitrogen, A10525) was used as a secondary antibody. The sections were stained with Wheat Germ Agglutinin, Alexa Fluor 488 Conjugate (Invitrogen, W11261, 1:10), and Hoechst (Life Technologies, 33258, 1:1000), and photographed with a confocal microscope (C2, Nicon). Three fields were randomly selected from each section for quantification.

## Microarray analysis

GSE28440, a microarray dataset testing RNA from interscapular brown adipose tissue was available in a public database (NCBI GEO: https://www.ncbi.nlm.nih.gov/geo/query/acc.cgi?acc=GSE28440).

## Statistical analysis

Statistical analysis was done with SPSS software (version 20) and figures were made with Prism9. Outliers and abnormal values were excluded by boxplot analyses for further analyses otherwise mentioned. Details are described in raw data files. In some cases, abnormal values were excluded from the description of the figure when they became out of range. At studies with qPCR, ΔCP showing values with an outlier or abnormal value in control group were excluded for further studies. SPSS uses Tukey's method to identify outliers or abnormal values and these are displayed on boxplots based on the following algorithm: Outlier (3rd quartile + 1.5 interquartile range, 1st quartile − 1.5 interquartile range), Abnormal value (3rd quartile + 3 interquartile range, 1st quartile − 3 interquartile range). Data are shown as the mean ± standard error of the mean (SEM). Differences between groups were examined by the independent-samples T-test or two-way analysis of variance (ANOVA), followed by Tukey's multiple comparison tests, the non-parametric Kruskal Wallis test, or the Dunnett's test for comparisons among more than two groups. Levene's test of equality of variances was used to test for homogeneity of variances and compared using an independent-samples T-test. The Welch's T-test was used in further analyses when the assumption of homogeneity of variances was violated and when the groups differed in the number of samples studied. A *P* value of *P* < 0.05 was considered statistically significant in all analyses.

# Data availability

GSE221854, a microarray dataset from BAT of adipose-specific vascular endothelial growth factor alpha (VEGFα) knockout or wild-type (WT) mice (12–14 weeks of age on regular chow) was analyzed by Boston University Microarray and Sequencing Resource (BUMSR) Core and have been deposited to NCBI GEO with the dataset identifier GSE221854. The source data of this paper are collected in the following database record: https://www.ebi.ac.uk/biostudies/studies/S-BSST1494. All materials and information can be requested to Ippei Shimizu (E-mail: shimizu.ippei@ncvc.go.jp).

The source data of this paper are collected in the following database record: biostudies:S-SCDT-10_1038-S44318-024-00196-0.

## Peer review information

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

## Acknowledgements

We thank Kaori Yoshida, Keiko Uchiyama, Satomi Kawai, Naomi Hatanaka, Yoko Sawaguchi, Runa Washio, Takako Ichihashi, Nanako Koike, Keiko Uchiyama, Masaaki Nameta (Niigata University), Emiko Nakamura, Nami Ozawa, Miki Ueda, Akane Masubuchi (Juntendo University), Tomomi Yamamoto, Yuki Shimomura, Manami Sone (National Cerebral and Cardiovascular Center Research Institute) for their excellent technical assistance, C. Ronald Kahn (Joslin Diabetes Center and Harvard Medical School) for providing the BAT cell line, Evan Rosen (Harvard Medical School) for providing us Ucp-Cre mice, Adam Gower (Boston University Microarray and Sequencing Resource (BUMSR) Core) for his technical support. This work was supported by Fusion Oriented Research for Disruptive Science and Technology (JST FOREST Program) (JPMJFR200L), JSPS Grants-in-Aid for Scientific Researcher (B) (23K27602), AMED Project for Elucidating and Controlling Mechanisms of Aging and Longevity under Grant Number JP21gm5010002, JSPS KAKENHI, Grant-in-Aid for Challenging Exploratory Research (17K19648), Grants-in-Aid for Encouragement of Young Scientists (A) (16H06244), Intramural Research Fund for Cardiovascular Diseases of the National Cerebral and Cardiovascular Center, Daiichi Sankyo's research grant (TaNeDS), Chugai Foundation for Innovative Drug Discovery Science, Foundation for Medical & Pharmaceutical Research, the Uehara Memorial Foundation, Kowa Life Science Foundation, Manpei Suzuki Diabetes Foundation, MSD Life Science Foundation, Public Interest Incorporated Foundation, Kanae Foundation Research Grant, Inamori Foundation, TERUMO FOUNDATION for LIFE SCIENCES and ARTS, SENSHIN Medical Research Foundation, ONO Medical Research Foundation, The Nakajima Foundation, SUZUKEN memorial foundation, HOKUTO Corporation, Mochida Memorial Foundation for Medical & Pharmaceutical Research, The Cell Science Research Foundation, Daiichi Sankyo Foundation of Life Science, Tokyo Biochemical research Foundation, Astellas Foundation for Research on Metabolic Disorders, Research grant from Naito Foundation, The Japan Geriatrics Society, Japan

Diabetes Foundation and Novo Nordisk Pharma Ltd., Kobayashi Magobei Memorial Medical Promotion Foundation of Japan, President's Grant for Interfaculty Collaboration Juntendo University, Japanese Circulation Society Grant for Future-Pioneering Doctors for Basic Research (to IS); by a Grant-in-Aid for Scientific Research (C) (22K08215), Daiichi Sankyo Foundation of Life Science, SENSHIN Medical Research Foundation, Astellas Foundation for Research on Metabolic Disorders, Japan Diabetes Foundation, Life Science Foundation of JAPAN (to YY), and by a grant from Bourbon (to TM and YY); by grants from JSPS Grant-in-Aid for Research Activity Start-up (23K19579) and Early-Career Scientists (24K18656), Intramural Research Fund for Cardiovascular Diseases (young researcher) of the National Cerebral and Cardiovascular Center, The Japan Foundation for Applied Enzymology, and Suzuken Memorial Foundation (to YTH).

## Author contributions

**Yung Ting Hsiao**: Investigation. **Yohko Yoshida**: Investigation. **Shujiro Okuda**: Investigation. **Manabu Abe**: Investigation. **Seiya Mizuno**: Resources. **Satoru Takahashi**: Resources. **Hironori Nakagami**: Resources. **Ryuichi Morishita**: Resources. **Kenya Kamimura**: Resources. **Shuji Terai**: Resources. **Tin May Aung**: Investigation. **Ji Li**: Investigation. **Takaaki Furihata**: Investigation. **Jing Yuan Tang**: Investigation. **Kenneth Walsh**: Resources. **Akihito Ishigami**: Investigation. **Tohru Minamino**: Investigation. **Ippei Shimizu**: Conceptualization; Supervision; Funding acquisition; Investigation; Methodology; Writing—original draft; Project administration; Writing—review and editing.

Source data underlying figure panels in this paper may have individual authorship assigned. Where available, figure panel/source data authorship is listed in the following database record: biostudies:S-SCDT-10_1038-S44318-024-00196-0.

## Disclosure and competing interests statement

The authors declare no competing interests.

# Expanded View Figures

**Figure EV1.  Related to Fig. 1.** ▶

C57BL/6NCrSlc wild-type (WT) mice were fed with a normal chow (NC) or a high-fat diet (HFD) from 4 weeks of age and studied at 38–40 weeks of age. (**A, B**) Body weight (BW) (**A**) ($n = 4, 6$) or brown adipose tissue (BAT) weight (**B**) ($n = 4, 5$) of indicated mice. (**C**) Reference Expression Dataset (RefEx) showing transcript *Pcolce* in systemic organs. (**D–J**) Gene Expression Omnibus studies showing transcript *Pcolce* in eWAT and BAT (**D**) ($n = 3, 3$ in GSE8044). The transcript was also tested under a HFD fed condition in mice in BAT (**E**) ($n = 3, 3$ in GSE64718), eWAT ($n = 4, 4$), liver ($n = 4, 4$), skeletal muscle (SM) (**F**) ($n = 3, 4$ in GSE123394), bone (**G**) ($n = 3, 4$ in GSE194075), adrenal gland (**H**) ($n = 4, 4$ in GSE216327), skin (**I**) ($n = 5, 5$ in GSE96932) and in heart (**J**) ($n = 4, 4$ in GSE171710). (**K–M**) Tabula Muris Senis testing transcript *Pcolce* in systemic organs (**K**), or in cells in the heart (**L**) or BAT (**M**). (**N–P**) Results from quantitative PCR (qPCR) showing transcript *Pcolce* in primary cardiac fibroblasts and primary brown adipocytes (**N**) ($n = 6, 6$), primary hepatocytes and primary brown adipocytes (**O**) ($n = 5, 5$), and hepatocyte (AML12) or brown adipocyte cell lines (**P**) ($n = 11, 12$). (**Q**) Relative fibrotic area (area/view) in the liver of indicated mice related to Fig. 1J ($n = 4, 5$). Data were analyzed by an independent-samples T-test. Data information: Representative data of two or more independent series (EV1A, B, N, P, Q), one independent series (EV1O). *$P < 0.05$, **$P < 0.01$. NS = not significant. Values represent the mean ± SEM. All data are from different biological replicates.  Source data are available online for this figure.

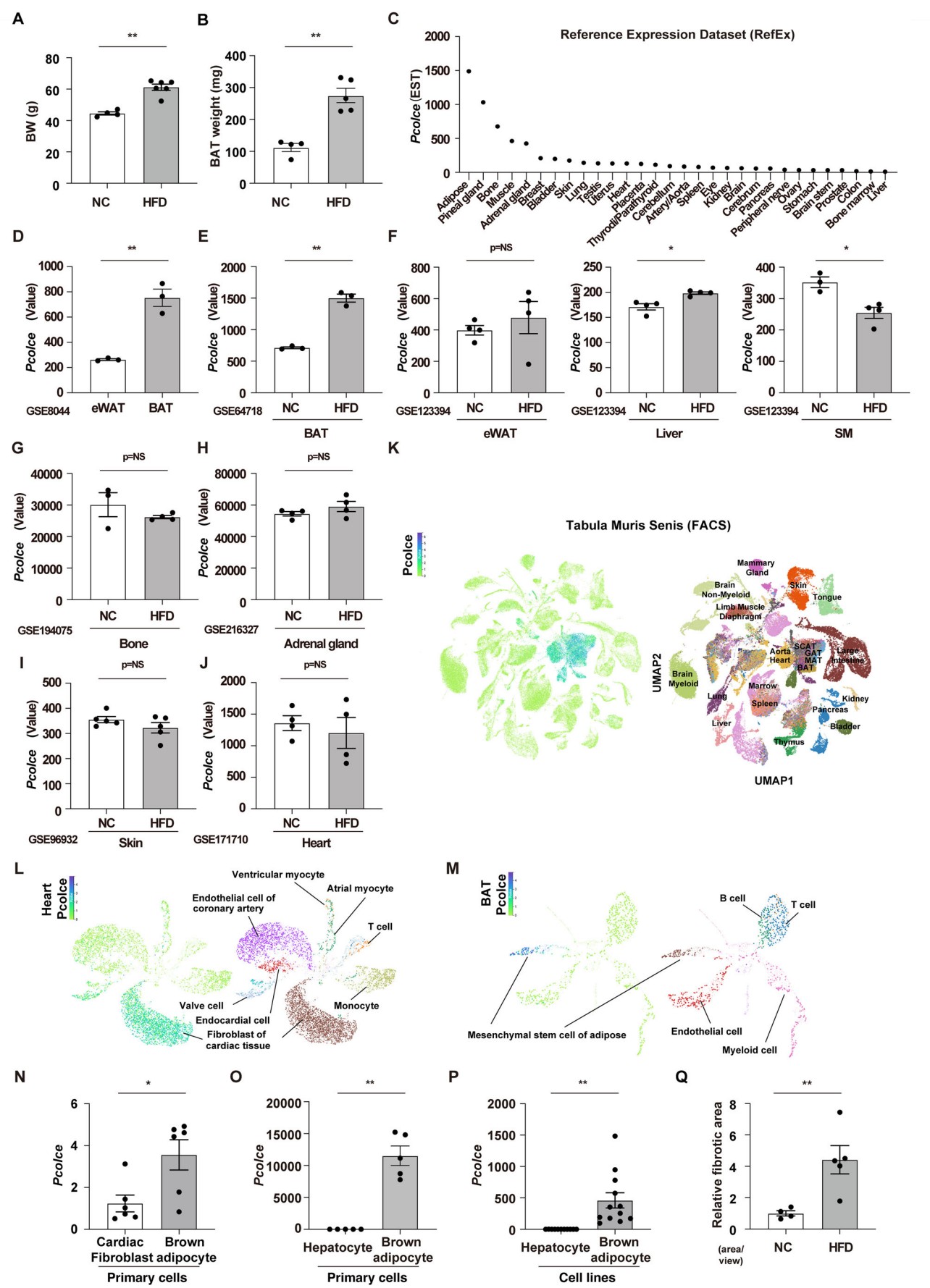

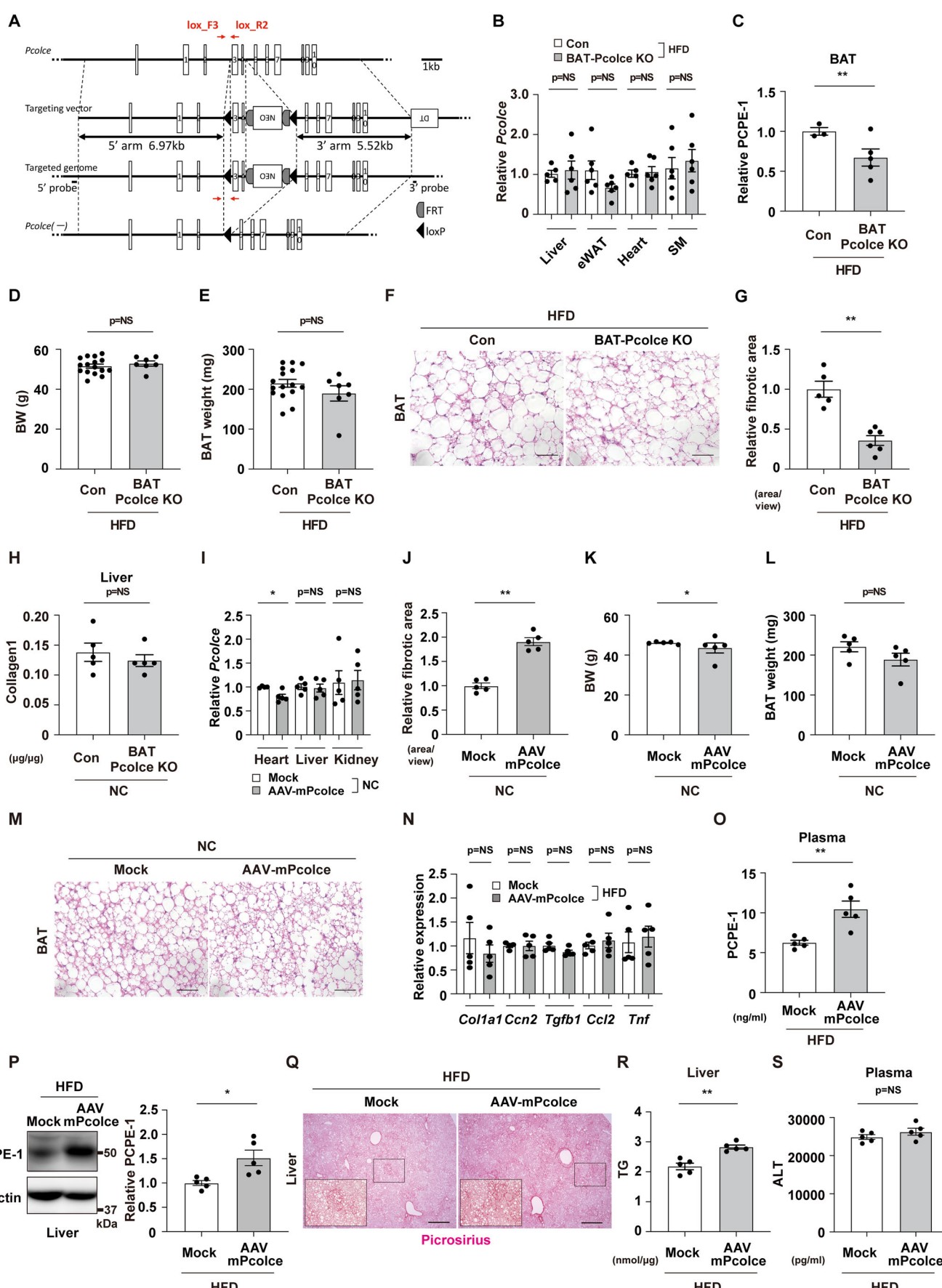

◀ **Figure EV2. Related to Fig. 2.**

Brown adipose tissue (BAT)-specific *Pcolce* knockout (KO) mice (UCP1-Cre$^{+/-}$; *Pcolce*$^{flox/flox}$; BAT *Pcolce* KO) or littermate control mice (UCP1-Cre$^{-/-}$; *Pcolce*$^{flox/flox}$; Con) were fed a high-fat diet (HFD) from 4 weeks of age and studied at 42–43 weeks of age. (A) Design of the BAT *Pcolce* KO mice. (B) Results from quantitative PCR (qPCR) showing transcript *Pcolce* in the liver ($n = 5, 6$), epididymal white adipose tissue (eWAT) ($n = 6, 6$), heart ($n = 5, 6$), and skeletal muscle (SM) (quadriceps) ($n = 6, 6$) from indicated mice fed a HFD. (C) The quantification of western blot analysis for PCPE-1 relative to Actin in BAT ($n = 3, 5$) related to Fig. 2B. (D, E) Body weight (BW) (D) ($n = 16, 7$), or BAT weight (E) ($n = 16, 7$) of indicated mice. (F) HE staining of the BAT from the indicated mice. Scale bar = 50 µm. (G) Relative fibrotic area in liver in the indicated mice, related to Fig. 2E ($n = 5, 6$). (H) ELISA for liver collagen type I (Collagen1) in the indicated mice ($n = 5, 5$). (I) Results from quantitative PCR (qPCR) showing transcript *Pcolce* in heart ($n = 4, 5$), liver ($n = 5, 5$) and kidney ($n = 5, 5$) in the indicated mice. (J) Relative fibrotic area of the liver in BAT AAV-Pcolce injection model (AAV-mPcolce) maintained on a normal chow (NC) related to Fig. 2N (58 weeks of age) ($n = 5, 5$). (K, L) Body weight (BW) (K) ($n = 5, 5$), or BAT weight (L) ($n = 5, 5$) of indicated mice. (M) HE staining of the BAT from the indicated mice. Scale bar = 50 µm. (N) Results from quantitative PCR (qPCR) showing transcripts *Col1a1* ($n = 5, 5$), *Ccn2*(Ctgf) ($n = 4, 5$), *Tgfb1* ($n = 5, 5$), *Ccl2* ($n = 5, 5$), and *Tnf* ($n = 5, 5$) in the liver from the indicated mice. (O) ELISA for PCPE-1 in plasma from the indicated mice ($n = 5, 5$). (P) Western blot analysis of PCPE-1 expression in liver from the indicated mice. The right panel indicates the quantification of PCPE-1 relative to Actin ($n = 5, 5$). (Q) Picrosirius red staining of liver from the indicated mice. Scale bar = 500 µm. The quantification results see Figure Appendix Figure S2H. (R, S) ELISA for liver triglyceride (TG) (R) ($n = 5, 5$) or plasma alanine transaminase (ALT) (S) ($n = 5, 5$) in the indicated mice. All data were analyzed by an independent-samples T-test. Data information: Representative data of two or more independent series (EV2B–E, G), one independent series (EV2I–L, N, P), one independent experiment analyzing samples from at least 2 independently prepared samples (EV2H, O, R, S). *$P < 0.05$, **$P < 0.01$. NS = not significant. Values represent the mean ± SEM. All data are from different biological replicates. Source data are available online for this figure.

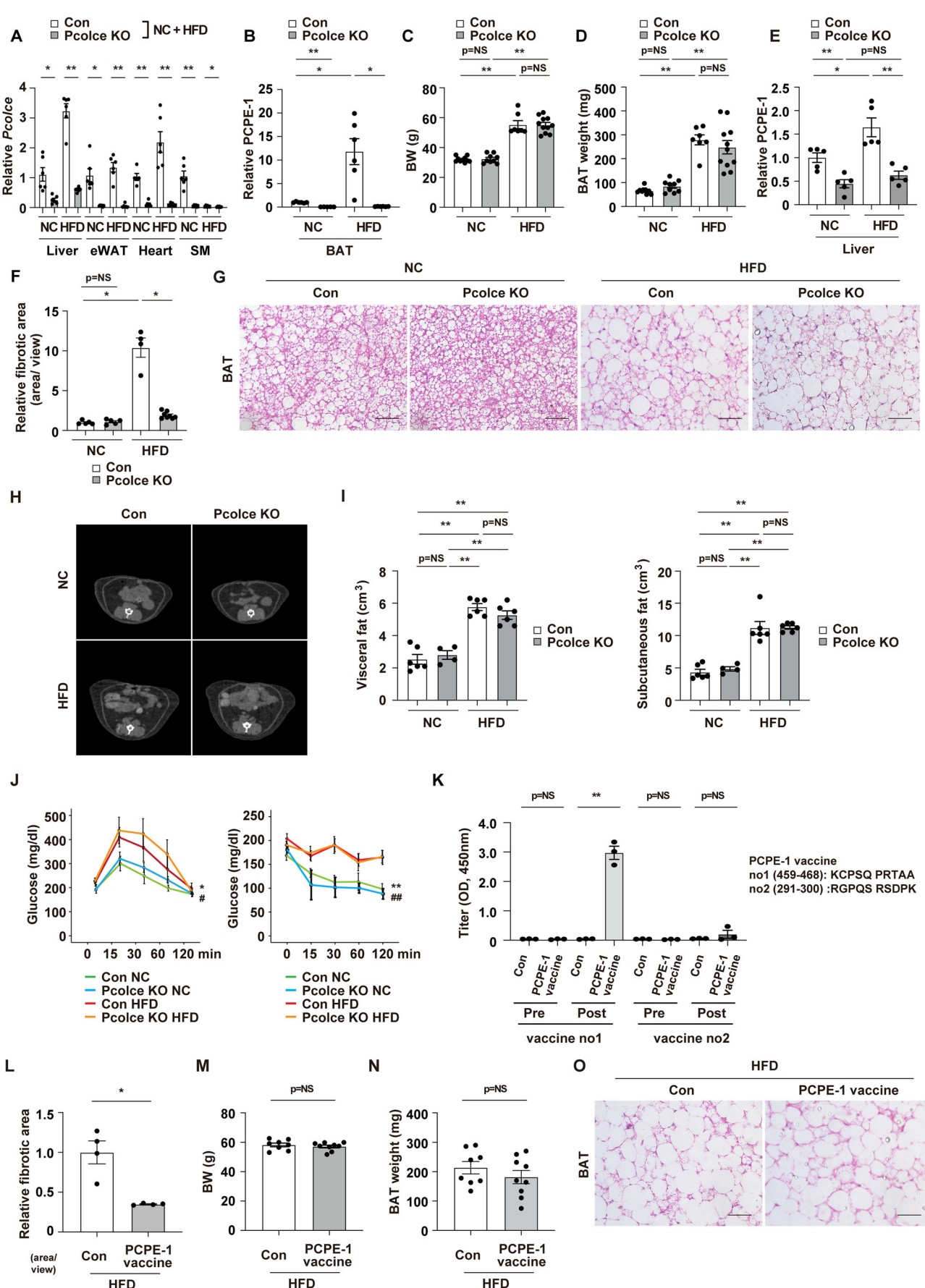

◄

**Figure EV3.  Related to Fig. 3.**

Systemic *Pcolce* knockout (*Pcolce* KO) or littermate control (Con) mice were maintained on a HFD from 4 weeks of age, and tissues were harvested and analyzed at 38–44 weeks of age. (**A**) Results from quantitative PCR (qPCR) showing transcript *Pcolce* in the liver, epididymal white adipose tissue (eWAT), heart, and skeletal muscle (SM) (quadriceps) from the indicated mice (heart: $n = 5, 6, 6, 6$, all other organs: $n = 6, 6, 6, 6$). (**B**) The quantification of western blot analysis for PCPE-1 relative to GAPDH in BAT related to Fig. 3B ($n = 5, 6, 6, 6$). (**C, D**) Body weight (BW) ($n = 10, 9, 7, 11$) (**C**), or BAT weight ($n = 10, 9, 7, 11$) (**D**) of indicated mice. (**E**) The quantification of western blot analysis for PCPE-1 relative to Actin in liver related to Fig. 3D ($n = 5, 5, 5, 5$). (**F**) Relative fibrotic area in the liver of indicated mice, related to Fig. 3E ($n = 5, 5, 4, 8$). (**G**) HE staining of the BAT from the indicated mice. Scale bar = 50 μm. (**H**) CT scan analyzing the abdominal area of indicated mice aged 29 weeks of age. (**I**) Volume of visceral or subcutaneous fat in the subdiaphragm abdominal area analyzed with a CT scan ($n = 6, 4, 6, 6$). (**J**) Glucose tolerance test (GTT) ($n = 6, 4, 8, 8$) or insulin tolerance test (ITT) ($n = 6, 4, 8, 7$) of the indicated mice. * and ** indicates Con NC vs Con HFD. # and ## indicates Pcolce KO NC vs Pcolce KO HFD. (**K**) The antibody titer in plasma was studied at pre- and post-immunization with two antigenic PCPE-1 peptide sequences (vaccine no.1 (KCPSQ PRTAA) and no.2 (RGPQS RSDPK)) ($n = 3, 3, 3, 3, 3, 3, 3$). (**L**) Relative fibrotic area in the liver of indicated mice, related to Fig. 3L ($n = 4, 4$). (**M, N**) Body weight (BW) (**M**) ($n = 8, 9$), and BAT weight (**N**) ($n = 8, 9$) of mice with or without PCPE-1 vaccination (vaccine no.1). (**O**) HE staining of the BAT from the indicated mice. Scale bar = 50 μm. All mice were maintained on a HFD and analyzed at 47 weeks age. The data in (**B–D**), (**F**), (**I**) were analyzed by a two-way analysis of variance (ANOVA) followed by Tukey's multiple comparison test (**I**), Dunnett test (**F**), or non-parametric Kruskal Wallis test (**C, D**). All the values in (**A**) were analyzed by a two-way ANOVA followed by Dunnett test except for NC in liver and HFD groups in SM analyzed by the independent-samples T-test. In (**B**), Con NC vs Con HFD was analyzed with the independent-samples T-test and other values were analyzed by a two-way ANOVA followed by Dunnett test. The data in (**J**) were analyzed with two-way repeated measures ANOVA. Other data were analyzed by an independent-samples T-test. Data information: Representative data of two or more independent series (EV3A–F, L–N), one independent series (EV3I–K). * and # $P < 0.05$, ** and ##$P < 0.01$. NS = not significant. Values represent the mean ± SEM. All data are from different biological replicates. Source data are available online for this figure.

**A**

Putative promotor region for Pcolce

| Transcriptional factor | Score | Relative score | Start | End | Strand | predicted site sequence |
|---|---|---|---|---|---|---|
| cJUN::cFOS | 5.682 | 0.818149714 | 15 | 21 | 1 | TGAAAGA |
| cJUN::cFOS | 6.687 | 0.854833858 | 215 | 221 | 1 | TGAAGCA |
| cJUN::cFOS | 6.574 | 0.850709173 | 304 | 310 | −1 | GGACTCA |
| cJUN::cFOS | 5.521 | 0.81227295 | 566 | 572 | −1 | TGCGTCA |
| cJUN::cFOS | 7.794 | 0.89524117 | 566 | 572 | 1 | TGACGCA |
| cJUN::cFOS | 8.185 | 0.90951331 | 600 | 606 | 1 | TGCCTCA |
| cJUN::cFOS | 6.656 | 0.853702308 | 719 | 725 | 1 | TGACTCC |
| cJUN::cFOS | 5.466 | 0.81026536 | 862 | 868 | −1 | GGAATCA |
| cJUN::cFOS | 6.656 | 0.853702308 | 974 | 980 | −1 | TGACTCC |
| cJUN::cFOS | 7.573 | 0.887174308 | 990 | 996 | −1 | TTAATCA |
| cJUN::cFOS | 6.789 | 0.858557025 | 1026 | 1032 | −1 | TGACAGA |
| cJUN::cFOS | 6.201 | 0.837094063 | 1185 | 1191 | 1 | TTCCTCA |
| cJUN::cFOS | 6.696 | 0.855162373 | 1189 | 1195 | −1 | TTACTGA |

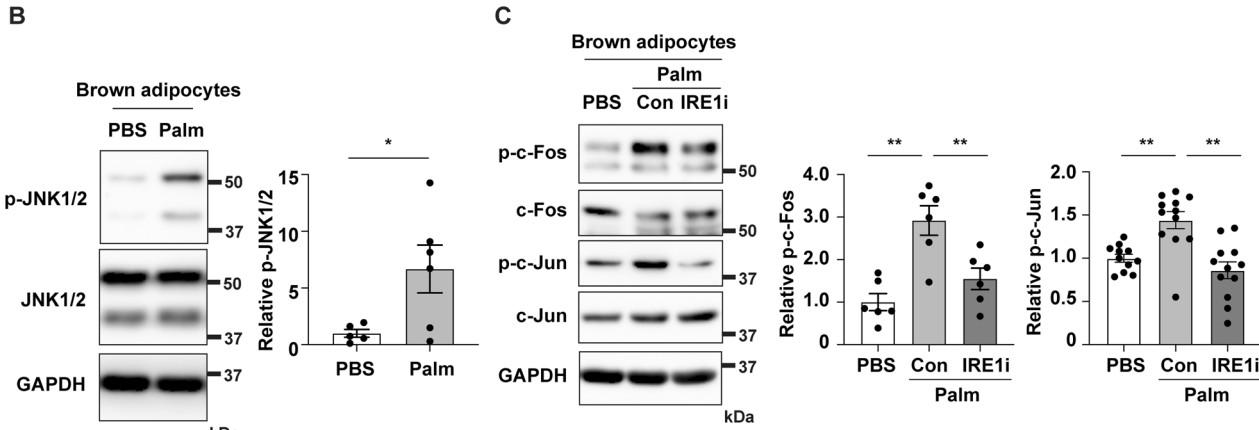

**B**  **C**

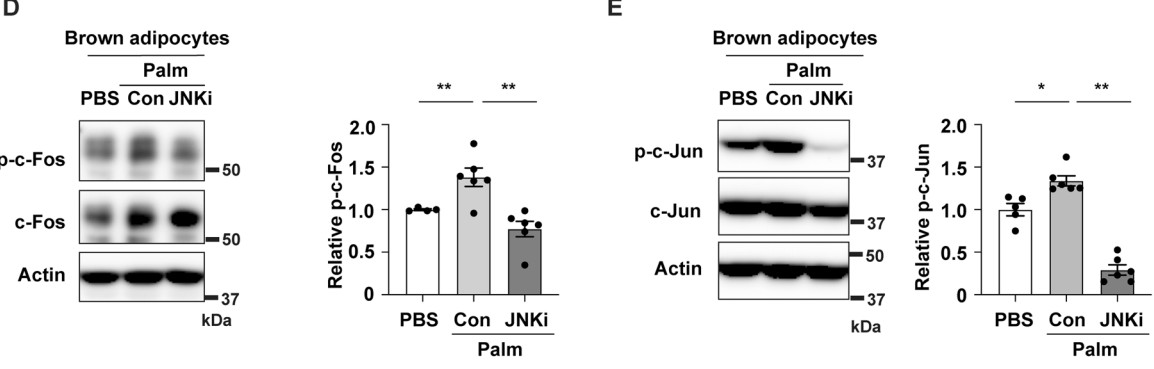

**D**  **E**

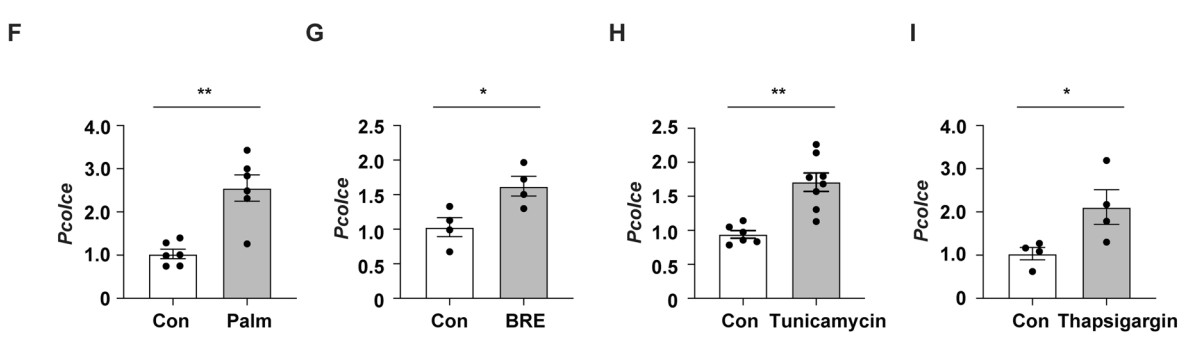

**F**  **G**  **H**  **I**

**Figure EV4. Related to Fig. 4.**

(A) Results of in silico analysis (DBTSS [http://dbtss.hgc.jp] and JASPAR [http://jaspar.binf.ku.dk]) predicting c-Fos and c-Jun may bind to putative promotor region of *Pcolce*. (B) Western blot analysis for phospho-JNK1/2 (p-JNK1/2) and JNK1/2 in fully differentiated brown adipocytes administrated with or without palmitic acid (Palm, 100 μM). The right panel indicates the quantification of p-JNK1/2 ($n = 6, 6$) relative to GAPDH. (C–E) Western blot analysis for p-c-Fos, c-Fos, p-c-Jun, and c-Jun in fully differentiated brown adipocytes administrated with PBS, palmitic acid (Palm; 100 μM for 6 h) with or without an IRE1 inhibitor (STF-083010, 30 μM, pre-treated 1 h) (C) or a JNK inhibitor (SP600125, 2 μM, pre-treated 1 h) (D, E). The right panels indicate the quantification of p-c-Fos and p-c-Jun relative to GAPDH or Actin (C: $n = 6, 6, 6$ for p-c-Fos, $n = 11, 12, 12$ for p-c-Jun) (D: $n = 4, 6, 6$) (E: $n = 5,6,6$). (F–I) Results from quantitative PCR (qPCR) showing transcript *Pcolce* in the differentiated brown adipocytes administrated with Palm (E) ($n = 6, 6$), BRE (Brefeldin A) (G) ($n = 4, 4$), Tunicamycin (H) ($n = 6, 8$) or Thapsigargin (I) ($n = 4, 4$). Data in Fig. EV4C–E were analyzed by a two-way analysis of variance (ANOVA) followed by Tukey's multiple comparison test. Other data were analyzed by an independent-samples T-test. Data information: Representative data of two or more independent series (EV4B–I). *$P < 0.05$, **$P < 0.01$. NS = not significant. Values represent the mean ± SEM. All data are from different biological replicates. Source data are available online for this figure.

