## [Peer Review File · The EMBO Journal]

PCPE-1, a brown adipose tissue-derived cytokine, promotes obesity-induced liver fibrosis

Ippei Shimizu, Yung Ting Hsiao, Yohko Yoshida, Shujiro Okuda, Manabu Abe, Seiya Mizuno, Satoru Takahashi, Hironori Nakagami, Ryuichi Morishita, Kenya Kamimura, Shuji Terai, Tin May Aung, Ji Li, Takaaki Furihata, Jing Yuan Tang, Kenneth Walsh, Akihito Ishigami, and Tohru Minamino

Corresponding author: Ippei Shimizu (shimizu.ippei@ncvc.go.jp)

Review Timeline:

Submission Date:	20th Oct 23
Editorial Decision:	28th Nov 23
Revision Received:	7th Mar 24
Editorial Decision:	6th Jul 24
Revision Received:	10th Jul 24
Accepted:	22nd Jul 24

Editor: Daniel Klimmeck

Transaction Report:

Dear Dr Shimizu,

Thank you again for the submission of your manuscript (EMBOJ-2023-115949) to The EMBO Journal. As mentioned earlier, your study was assessed by two reviewers with expertise in adipocyte biology and systemic metabolism, whose comments are enclosed below.

As you will see from their comments, the referees acknowledge the analysis and potential interest and value of your findings. However, they also express major concerns, which need to be addressed thoroughly to make them supportive of publication in the EMBO Journal. In more detail, referee #1 raises important issues about the specificity of PCPE1 expression in BAT (ref#1, pt.2), as well as the depth of mechanistic insights on control of PCPE-1 by an interplay between AP-1 transcription factors and JNK (ref#1, pt.4). This point is echoed by reviewer #2 who states that both upstream regulation of PCPE-1 and its downstream effectors are too prematurely explored at this point (ref#2, pts. 1,7). This expert also points to preceding literature already linking PCPE-1 to liver fibrosis (ref#2, pt.2). Further, the reviewers raise a number of issues related to the presentation of the findings, additional controls and improved methods annotation required, statistics applied and overall discussion of related literature, that would need to be conclusively addressed to achieve the level of robustness and clarity needed for The EMBO Journal.

Given the overall interest stated and broader angle of your findings, we are able to invite you to revise your manuscript experimentally to address the referees' comments. I need to stress though that we do require strong support from the referees on a revised version of the study in order to move on to publication of the work, and i.p. the mechanistic concerns would need to be addressed in order to make this a compelling case to move forward with for the EMBO Journal.

In light of the extensive experimentation requested, I would appreciate if you could contact me during the next weeks for exchange e.g. a video call to discuss your perspective on the comments and potential plan for revisions.

Please feel free to contact me if you have any questions or need further input on the referee comments.

When submitting your revised manuscript, please carefully review the instructions below.

Please feel free to approach me any time should you have additional questions related to this.

Thank you for the opportunity to consider your work for publication.

I look forward to your revision.

Kind regards,

Daniel Klimmeck

Daniel Klimmeck, PhD
Senior Editor
The EMBO Journal

Instruction for the preparation of your revised manuscript:

2) individual production quality figure files as .eps, .tif, .jpg (one file per figure).

3) a .docx formatted letter INCLUDING the reviewers' reports and your detailed point-by-point response to their comments. As part of the EMBO Press transparent editorial process, the point-by-point response is part of the Review Process File (RPF),

which will be published alongside your paper.

4) a complete author checklist, which you can download from our author guidelines ([https://wol-prod-cdn.literatumonline.com/pb-assets/embo-site/Author Checklist%20-%20EMBO%20J-1561436015657.xlsx](https://wol-prod-cdn.literatumonline.com/pb-assets/embo-site/Author%20Checklist%20-%20EMBO%20J-1561436015657.xlsx)). Please insert information in the checklist that is also reflected in the manuscript. The completed author checklist will also be part of the RPF.

6) It is mandatory to include a 'Data Availability' section after the Materials and Methods. Before submitting your revision, primary datasets produced in this study need to be deposited in an appropriate public database, and the accession numbers and database listed under 'Data Availability'. Please remember to provide a reviewer password if the datasets are not yet public (see <https://www.embopress.org/page/journal/14602075/authorguide#datadeposition>).

7) Our journal encourages inclusion of *data citations in the reference list* to directly cite datasets that were re-used and obtained from public databases. Data citations in the article text are distinct from normal bibliographical citations and should directly link to the database records from which the data can be accessed. In the main text, data citations are formatted as follows: "Data ref: Smith et al, 2001" or "Data ref: NCBI Sequence Read Archive PRJNA342805, 2017". In the Reference list, data citations must be labeled with "[DATASET]". A data reference must provide the database name, accession number/identifiers and a resolvable link to the landing page from which the data can be accessed at the end of the reference. Further instructions are available at .

8) At EMBO Press we ask authors to provide source data for the main and EV figures. Our source data coordinator will contact you to discuss which figure panels we would need source data for and will also provide you with helpful tips on how to upload and organize the files.

Numerical data can be provided as individual .xls or .csv files (including a tab describing the data). For 'blots' or microscopy, uncropped images should be submitted (using a zip archive or a single pdf per main figure if multiple images need to be supplied for one panel). Additional information on source data and instruction on how to label the files are available at .

9) We replaced Supplementary Information with Expanded View (EV) Figures and Tables that are collapsible/expandable online (see examples in <https://www.embopress.org/doi/10.15252/emboj.201695874>). A maximum of 5 EV Figures can be typeset. EV Figures should be cited as 'Figure EV1, Figure EV2' etc. in the text and their respective legends should be included in the main text after the legends of regular figures.

11) For data quantification: please specify the name of the statistical test used to generate error bars and P values, the number (n) of independent experiments (specify technical or biological replicates) underlying each data point and the test used to calculate p-values in each figure legend. The figure legends should contain a basic description of n, P and the test applied. Graphs must include a description of the bars and the error bars (s.d., s.e.m.).

Please remember: Digital image enhancement is acceptable practice, as long as it accurately represents the original data and

conforms to community standards. If a figure has been subjected to significant electronic manipulation, this must be noted in the figure legend or in the 'Materials and Methods' section. The editors reserve the right to request original versions of figures and the original images that were used to assemble the figure.

We realize that it is difficult to revise to a specific deadline. In the interest of protecting the conceptual advance provided by the work, we recommend a revision within 3 months (26th Feb 2024). Please discuss the revision progress ahead of this time with the editor if you require more time to complete the revisions.

Referee #1:

The work by Hsiao et al. demonstrates a potential therapeutic in suppressing brown fat-derived procollagen C-endopeptidase enhancer (PCPE-1) for treating nonalcoholic fatty liver disease in a mouse model. The work presented shows new results on targeting PCPE-1 and ameliorating NAFLD development. The authors use several different challenging methods to substantiate their findings and aim to provide a mechanistic understanding of PCPE-1 in NAFLD. However, there are several points that are unclear as follows:

Major points:

1. There seems to be a statistical analysis, but how is this being achieved is unclear. The authors show multiple graphics with large SEM bars using a total n of 5 vs 5 and claiming to have p value below 0.05. The authors throughout the manuscript are analyzing results using the two-tailed Student's t-test for differences between two groups. While this is adequate, in some cases there are situations where the authors compare 9 vs 19. This is in fact an incorrect way to analyze the data as it will give rise to false discoveries. It would be extremely relevant for the authors to re-analyze the data where the sample size is unequal and do the Welch's t-test, in order to account for this difference in sample size.
2. Figure 1c shows copy numbers/ μ l. This is not normalized to anything, which makes the results incomparable. It has been shown that PCPE-1 expression is high in many Collagen-rich tissues, such as muscle, bone, and skin. This casts doubt on PCPE-1 being predominantly expressed in BAT. How is the regulation in cultured brown adipocytes?
3. The authors claim to have generated a BAT specific PCPE-1 gain of function model. However, the serotype of AAV used is not specified, as well as there are no proof of specificity. Furthermore, the promoter used in the vector is not tissue specific. To claim specificity the authors should at least show that PCPE-1 expression remains unaltered in other organs, after AAV injection.
4. The authors claim to have elucidated the mechanism behind upregulation of PCPE-1. However, there is some important results missing to make this claim. The authors show that overexpressing c-Fos and c-Jun together result in a small increment of PCPE-1, yet in the following up experiments of activating c-Fos and c-Jun with either brefeldin or palmitic acid, there is no evidence that this promotes PCPE-1 expression as hypothesized. As such it would be important to show that indeed causing ER stress does lead to a significant increment in PCPE-1 expression and in BAT it has been shown to be linked to proteasome function (PMID: 29400713). Please use proteasome inhibitors like bortezomib, tunicamycin, and thapsigargin in cell culture to support your claims.

Minor points:

5. The authors use total amount of ALT produced in liver to assess for liver damage. This is a poor choice to indicate liver damage, as in order to assess liver damage it is relevant to assess ALT (and AST and the ratio) present in plasma. This way of measurement gives rise to conflicting results as in figure 1 panel i wildtype animals under chow diet show levels of ALT in the liver from 60 to 100 pg/ g. However, in figure 2 panel p the authors show that wildtype animals have an ALT amount in the liver of around 600 pg/ g.
6. Data visualization is unclear using box plots. Please use bar plot graphs with individual data points for the animals.
7. While in the manuscript the histological staining is written properly as picrosirius red. In the figures the histological staining is named as picro silius, which is incorrect.
8. The cohorts of the human study are not matched by weight, which undermines the conclusion that PCPE-1 levels are a result of NASH. Furthermore, the size difference between groups would make a different kind of statistical analysis necessary, as previously described.
9. The Western Blot in 4a shows the presence of c-fos in NC BAT. This is entirely absent in the Mock-vector injected NC mice. This may indicate off-target effects of the vector.
10. The method by which you identify outliers is unclear from the methods.
11. The use of NAFLD or NASH is outdated, more updated are MAFLD and MASH

12. The authors show multiple western blots for PCPE-1. Interestingly, there is a great variability in PCPE-1 amount in between WT animals in chow diet. As it can be observed in figure 1 panel e and figure 2 panel k. The same can be said for expression of WT under HFD in figure 1 panel e, figure 2 panel b, and figure 3 panel b. Please include all uncropped immunoblots in the supplement. Also, you seem to infer statistical analysis, but I can only see single lane immunoblots - include all blots that have led to these statistics.

13. It is phospho, not phosphor - autocorrection it seems

Additional comment: It end it remains unclear what the physiological role of PCPE-1 is. Why is it produced in brown adipocytes? Only the pathological connection to ER stress, obesity, and MASH is highlighted here.

Referee #2:

In this study, the authors identified procollagen C-endopeptidase enhancer (PCPE-1) as a BATokine. The data shows that PCPE-1 is induced in brown adipose tissue (BAT) in response to a NASH model (prolonged high-fat feeding) and is secreted as BATokine, promoting fibrosis in mice. Either BAT-specific or global deletion of PCPE-1 prevented liver fibrosis in this model. In contrast, PCPE-1 overexpression in BAT was sufficient to promote liver fibrosis in chow-fed mice. The authors propose that HFD-induced ER stress in BAT leads to the activation of IRE1/JNK/c-Fos/c-Jun signaling, thereby inducing PCPE-1. The idea that PCPE-1 is induced in the context of liver fibrosis and may contribute to its development is not new, but this is the first time PCPE-1 is reported as a BATokine. The data suggest that BAT-liver communication mediated by PCPE-1 is critical for obesity-induced liver fibrosis. Although of interest, there are concerns regarding the data presented that impact the conclusions that can be drawn. For example, questions remain regarding the mechanisms mediating PCPE-1 induction in BAT, and regarding whether liver PCPE-1 levels itself also contribute to the development of fibrosis. Finally, although PCPE-1 could contribute to the development of liver fibrosis, other aspects of NASH were not ameliorated by PCPE-1 deletion, reducing the significance of these findings as a therapeutic strategy for the treatment of NASH.

Specific Comments

- 1) Although prolonged HFD induced high PCPE-1 levels in BAT, it also significantly induced PCPE-1 levels in the liver. Therefore, it is possible that liver PCPE-1 induction is sufficient to promote fibrosis in this obesity-induced NASH model. Several questions remain to be addressed: 1. Are PCPE-1 levels in liver affected by PCPE-1 deletion and/or overexpression in BAT? 2. Is liver PCPE-1 induction also required to mediate fibrosis in this model? 3. Would conditional deletion of PCPE-1 in the liver result in a similar phenotype as in BAT KO mice? Also, would overexpressing PCPE-1 in the liver be sufficient to induce fibrosis?
- 2) High PCPE-1 levels have been associated with liver fibrosis in previous studies and have been suggested as a potential biomarker and therapeutic target (PMCID: PMC49614444; PMC8377038; PMC5602882). Indeed, global PCPE-1 KO reduced liver fibrosis, but not NASH progression in mice (PMC8836302). The present study should be discussed in light of these initial observations. These studies should be cited and discussed in the present manuscript.
- 3) Serum levels of PCPE-1 are highly variable in the different settings described. In figure 1F PCPE1 levels in NC controls is 100ng/ml rising to ~ 275ng/ml with HFD. Levels in the AAV studies are 10ng/ml in the mock rising to 25 ng/ml in the AAV. These levels are lower than the normal range in figure 1, yet authors are invoking that this is sufficient to induce fibrosis. This is somewhat confusing and not consistent with a dose-response effect. Moreover, circulating levels of PCPE1 in humans are 2ng/ml. How much circulating PCPE-1 is required to elicit liver fibrosis? Why are these levels so variable? Does feeding status or any metabolic changes also affect PCPE-1 circulating levels?
- 4) There is an overall lack of characterization on BAT function and morphology in BAT PCPE-1 KO mice or in the overexpression model. Does BAT-derived PCPE-1 play any autocrine or paracrine roles that could potentially contribute to the liver phenotype?
- 5) Although body weight was unchanged between WT and PCPE-1 BAT KO mice fed HFD, it is unclear whether any other metabolic parameters were affected by PCPE-1 manipulation such as glucose homeostasis and body composition.
- 6) Several other genes related to fibrosis were identified to be induced in BAT in response to HFD, including Cyr61, which has anti-fibrotic properties. Is the expression of these genes altered upon PCPE-1 manipulation in BAT?
- 7) Although AAV-mediated activation of c-Fos and c-Jun is sufficient to induce PCPE-1 in brown adipocytes, the link with ER stress and IRE1 signaling and the requirement of c-Fos and c-Jun for PCPE-1 induction were not rigorously tested. Therefore, regarding the mechanisms for PCPE-1 induction in BAT, there are several issues that need to be addressed:
 1. Does inhibition of IRE1 and JNK prevent PCPE-1 induction in and secretion from brown adipocytes?
 2. Does deletion of c-Fos or c-Jun prevent PCPE-1 induction?
 3. Does brefeldin (BRE) treatment induce PCPE-1?

4. Can induction of PCPE-1 in BAT of obese animals or in brown adipocytes in vitro, be prevented by global attenuation of ER stress with a chemical chaperone, such as TUDCA?
5. Would activation of ER stress with more classical compounds such as tunicamycin and thapsigargin promote PCPE-1 induction in brown adipocytes?
6. The investigators have not demonstrated in vivo that inhibiting ER stress is sufficient to prevent the secretion of PCPE1. This could be achieved by treating animals with the chemical chaperone TUDCA.

- 8) Many of the data panels lack requisite controls. See for example Figures 2e, and all data panels in figure 3 should include data from normal chow-fed controls.

- 9) It is curious that no plasma concentrations of PCPE-1 are shown in control and HFD mice in the germline KO experiments depicted in Figure 3.

- 10) It is intriguing that BAT Pcoclce KO mice also have reduced TG. Why is this? And how does this contribute to the amelioration of the fibrosis phenotype?

- 11) Would HFD exacerbate the findings in the AAV mouse? It seems like the degree of fibrosis observed under in these mice under NC conditions is significantly less. They do seem to have a slightly more hepatic TG although there is no inflammation. This again raises the question regarding whether or not PCPE-1 could have pleiotropic effects.

Comments on the organization of the manuscript

1. As written the manuscript is not easy to follow. The authors include too much background and discussion in the presentation of the results. Some of these points should either be briefly consolidated in the introduction or moved to the discussion section.
2. The authors might consider reorganizing the flow of the data by initially focusing on the regulation of PCPE-1 in BAT prior to describing its putative role in the regulation of hepatic fibrosis.
3. The manuscript should be edited for grammar and some typos fixed such as Picrosirius in the figures... which is incorrectly labeled as Picro Sillius in figs 1j, 2d,m, 3d,j.
4. The terms Non-Alcoholic Steatohepatitis (NASH) and Non-Alcoholic Fatty Liver Disease (NAFLD) have been replaced Metabolic Dysfunction Associated Steatohepatitis (MASH) and Metabolic Dysfunction Associated Fatty Liver Disease (MAFLD). The terminology used in the manuscript should be replaced with this more contemporary nomenclature.

Response to the Editor:

We thank the Editor for giving us the opportunity to revise our manuscript. We performed additional experiments to address issues raised by the Reviewers.

Referee #1:

The work by Hsiao et al. demonstrates a potential therapeutic in suppressing brown fat-derived procollagen C-endopeptidase enhancer (PCPE-1) for treating nonalcoholic fatty liver disease in a mouse model. The work presented shows new results on targeting PCPE-1 and ameliorating NAFLD development. The authors use several different challenging methods to substantiate their findings and aim to provide a mechanistic understanding of PCPE-1 in NAFLD. However, there are several points that are unclear as follows:

Response:

We thank the reviewer for these helpful comments. We performed additional experiments to address the comments by the reviewer.

Major points:

1. There seems to be a statistical analysis, but how is this being achieved is unclear. The authors show multiple graphics with large SEM bars using a total n of 5 vs 5 and claiming to have p value below 0.05. The authors throughout the manuscript are analyzing results using the two-tailed Student's t-test for differences between two groups. While this is adequate, in some cases there are situations where the authors compare 9 vs 19. This is in fact an incorrect way to analyze the data as it will give rise to false discoveries. It would be extremely relevant for the authors to re-analyze the data where the sample size is unequal and do the Welch's t-test, in order to account for this difference in sample size.

Response:

We thank the reviewer for this criticism. When performing statistical analyses in the initial submission using the T test mode in the SPSS, which we found was an independent-samples T-test (not a Student *t* test), Levene's test of equality of variances was used to test the samples for homogeneity of variances, and the Welch's T-test was used in further analyses to test when the assumption of homogeneity of variances was violated. Based on the reviewer's suggestion,

however, we re-analyzed the studies involving different sample numbers using the Welch's T-test and found no difference in experimental results from those obtained in the initial submission. In the revised manuscript, therefore, we have included the following description in the Methods section: "Differences between groups were examined by the independent-samples T-test or two-way analysis of variance (ANOVA), followed by Tukey's multiple comparison tests, the non-parametric Kruskal Wallis test, or the Dunnett's test for comparisons between more than two groups. Levene's test of equality of variances was used to test for homogeneity of variances and compared using an independent-samples T-test. The Welch's T-test was used in further analyses when the assumption of homogeneity of variances was violated and when the groups differed in the number of samples studied." We hope this addresses the reviewer's concern.

2. Figure 1c shows copy numbers/ μ l. This is not normalized to anything, which makes the results incomparable. It has been shown that PCPE-1 expression is high in many Collagen-rich tissues, such as muscle, bone, and skin. This casts doubt on PCPE-1 being predominantly expressed in BAT. How is the regulation in cultured brown adipocytes?

Response:

In response, we have relabeled the unit as copy/ μ gRNA to make it more understandable. We checked the Reference Expression Dataset (RefEx) (<https://refex.dbcls.jp/index.php?lang=en>) (<https://www.nature.com/articles/sdata2017105>) and used the Expressed sequence tag (EST) obtained from the EST division of the INSD (International Nucleotide Sequence Database, consisting of Genbank/European Nucleotide Archive (ENA)/ DNA DataBank of Japan(DDBJ)). We found that the transcript *Pcolce* to be most highly expressed in adipose tissue, followed by the pineal gland, bone, muscle, adrenal gland. In the Gene Expression Omnibus (<https://www.ncbi.nlm.nih.gov/geo/>), GSE8044 demonstrated that the transcript *Pcolce* was more highly expressed (by about 2.5-3-fold) in BAT than in epididymal WAT (eWAT). We also examined GSE64718, GSE123394, GSE194075, GSE216327, GSE96932 and GSE171710 for *Pcolce* expression in obesity using BAT, eWAT, liver, skeletal muscle (SM), bone, adrenal gland and heart. We demonstrated that *Pcolce* was increased significantly not only in BAT (by about 2-fold) but in the liver and other organs, albeit to a much lesser degree (eWAT, bone, adrenal gland, skin and heart) or reduced (SM), with these results shown at the next page as a new figure (Fig. EV1C-J).

Fig EV1.

Next, we analyzed single-cell RNAseq data from the Tabula Muris Senis (<https://tabula-muris-senis.ds.czbiohub.org>) and found that *Pcolce* was predominantly expressed in BAT, WAT, aorta and heart (new Fig.EV1K-M (next page)). As already noted, GSE8044 demonstrated that the transcript *Pcolce* was more highly expressed (by about 2.5-3-fold) in BAT than in epididymal WAT (eWAT); again, *Pcolce* was most highly expressed in the fibroblasts and mesenchymal stem cells of adipose tissue in the heart and BAT, respectively.

Fig EV1.

We performed additional experiments in neonatal rats and demonstrated that the transcript *Pcolce* was more highly expressed (2.5-3-fold) in their brown adipocytes than in their fibroblasts, with these results shown in a new figure below (Fig. EV1N).

Fig EV1N.

We also tested the level of expression of the transcript *Pcolce* not only in primary hepatocyte and primary brown adipocytes collected from adult mice but in a hepatocyte cell line (AML12) and a brown adipocyte cell line (a gift from Dr. C. Ronald Kahn) and demonstrated that the level of expression of the transcript *Pcolce* was more than 10,000-fold higher in the primary brown adipocytes, and 500-fold higher in the brown adipocyte cell line, than in their respective hepatocytes, with these results shown in a new figure below at the next page (Figs. EV1O/P).

Fig EV10**Fig EV1P**
These data appear to indicate that brown adipose tissue and brown adipocytes likely represent a predominant source for PCPE-1 production.

Further, to examine PCPE-1 for its regulation in cultured brown adipocytes, we examined the transcript *Pcolce* for its level of expression before and after their differentiation, and found that these levels were comparable between the groups compared. Adrenergic signaling has a central role in the maintenance of homeostasis in brown adipocytes, however, administration of catecholamine to differentiated brown adipocytes produced no change in PCPE-1 either at its transcript level or at its protein level. Our data suggest that the ER stress/cFos/cJun pathway has critical roles in the regulation of *Pcolce* expression in brown adipocytes under metabolically stressed conditions. Mechanisms including epigenetic modification may be involved in the regulation of this molecule under physiological conditions, and we are hoping to test this hypothesis in our future studies.

3. The authors claim to have generated a BAT specific PCPE-1 gain of function model. However, the serotype of AAV used is not specified, as well as there are no proof of specificity. Furthermore, the promoter used in the vector is not tissue specific. To claim specificity the authors should at

least show that *PCPE-1* expression remains unaltered in other organs, after AAV injection.

Response:

We are grateful for these helpful comments. As per the reviewer's suggestion, we performed qPCR studies and found that the AAV-mPcolce model did not increase the level of the transcript *Pcolce* in other organs including the heart, liver and kidney. Thus, these results have been included as a new figure shown below (Fig. EV2I) in our revised manuscript.

4. The authors claim to have elucidated the mechanism behind upregulation of *PCPE-1*. However, there is some important results missing to make this claim. The authors show that overexpressing *c-Fos* and *c-Jun* together result in a small increment of *PCPE-1*, yet in the following up experiments of activating *c-Fos* and *c-Jun* with either brefeldin or palmitic acid, there is no evidence that this promotes *PCPE-1* expression has hypothesized. As such it would be important to show that indeed causing ER stress does lead to a significant increment in *PCPE-1* expression and in BAT it has been shown to be linked to proteasome function (PMID: 29400713). Please use proteasome inhibitors like bortezomib, tunicamycin, and thapsigargin in cell culture to support your claims.

Response:

We greatly appreciate these valuable comments and suggestions. In response, we went on to administer brefeldin (BRE), tunicamycin, thapsigargin, palmitic acid (Palm) to differentiated brown adipocytes, and demonstrated an increase in the level of the transcript *Pcolce* in these cells. Interestingly, we also found that this increase was not seen with the administration of bortezomib

(BTZ). We have therefore included these data for BRE, tunicamycin, thapsigargin and Palm as new figures shown below (Figs. EV4F-I).

Minor points:

5. The authors use total amount of ALT produced in liver to assess for liver damage. This is a poor choice to indicate liver damage, as in order to assess liver damage it is relevant to assess ALT (and AST and the ratio) present in plasma. This way of measurement gives rise to conflicting results as in figure 1 panel i wildtype animals under chow diet show levels of ALT in the liver from 60 to 100 pg/ μ g. However, in figure 2 panel p the authors show that wildtype animals have an ALT amount in the liver of around 600 pg/ μ g.

Response:

We are grateful for these insightful comments. In response, we re-analyzed plasma ALT levels in our models using ELISA and demonstrated that dietary obesity led to an increase in plasma ALT levels but that PCPE-1 manipulation did not affect this increase, with these results included as new figures shown below at the next page (Figs. 1I, 2H, 2Q, 3H, 3O, and EV2S).

6. Data visualization is unclear using box plots. Please use bar plot graphs with individual data points for the animals.

Response:

We apologize for the inconvenience caused. We have now revised our figures as suggested.

7. While in the manuscript the histological staining is written properly as picosirius red. In the figures the histological staining is named as picro silius, which is incorrect.

Response:

We apologize for this oversight. We have now revised our figures as suggested.

8. The cohorts of the human study are not matched by weight, which undermines the conclusion that PCPE-1 levels are a result of NASH. Furthermore, the size difference between groups would make a different kind of statistical analysis necessary, as previously described.

Response:

We thank the reviewer for raising this point, We have performed BMI-matched analyses testing

for PCPE-1 levels in the serum. Under this condition, we demonstrated that patients with MASH had significantly increased PCPE-1 levels. These data have been included as new figures (Appendix Figs. S1A/B).

Appendix Fig S1A

Appendix Fig S1B

9. The Western Blot in 4a shows the presence of c-fos in NC BAT. This is entirely absent in the Mock-vector injected NC mice. This may indicate off-target effects of the vector.

Response:

We thank the reviewer for bringing this issue to our attention. In response, we have performed a new western blot analysis with the group as shown in Fig. 4B, and confirmed a very weak signal in the Mock-vector group, despite its long exposure, through imaging studies. Given that this was consistent with the results shown in another research project at our lab, demonstrating that c-Fos signaling increases with aging in BAT, we would like to retain these panels as they were in the initial submission. We would also like to note that the mice studied in Fig. 4A were 32-38 weeks of age, while those studied in Fig. 4B were 13 weeks of age.

10. The method by which you identify outliers is unclear from the methods.

Response:

We are grateful to the reviewer for raising this. In response, we have checked the algorithm used in SPSS for detecting outliers or abnormal values and included the following statement in the Methods section: “SPSS uses Tukey’s method to identify outliers or abnormal values and these are displayed on boxplots based on the following algorithm: outlier (3rd quartile + 1.5 interquartile range, 1st quartile – 1.5 interquartile range), abnormal value (3rd quartile + 3 interquartile range, 1st quartile – 3 interquartile range).”

11. The use of NAFLD or NASH is outdated, more updated are MAFLD and MASH

Response:

We thank the reviewer for this helpful comment and have revised the description in the text.

12. The authors show multiple western blots for PCPE-1. Interestingly, there is a great variability in PCPE-1 amount in between WT animals in chow diet. As it can be observed in figure 1 panel e and figure 2 panel k. The same can be said for expression of WT under HFD in figure 1 panel e, figure 2 panel b, and figure 3 panel b. Please include all uncropped immunoblots in the supplement. Also, you seem to infer statistical analysis, but I can only see single lane immunoblots - include all blots that have led to these statistics.

Response:

We thank the reviewer for raising these issues. In response, we have optimized the protein concentration as 40 µg/sample and performed additional studies with the imager mode altered in some panels (i.e., Figs. 1E, 2B, 3B below). We have provided the full blot panels used for quantification as a raw data file.

13. *It is phospho, not phosphor - autocorrection it seems*

Response:

We are grateful for raising this. We have now corrected the description.

Additional comment: It end it remains unclear what the physiological role of PCPE-1 is. Why is it produced in brown adipocytes? Only the pathological connection to ER stress, obesity, and MASH is highlighted here.

Response:

We are grateful for this helpful comment. While we have confined ourselves to testing PCPE-1 for its pathogenic role in obesity focusing on MASH in our revised manuscript, another ongoing research at our lab also indicates that PCPE-1 increases with aging in humans and mice, and pilot studies also suggest that this molecule increases in patients with heart failure with preserved ejection fraction, chronic kidney disease and atrial fibrillation. Now our research group is on its way toward establishing the concept, “Age-related fibrotic disorders”, as one syndrome encompassing these diseases, with this concept received a Catalyst Award from National Academy of Medicine, USA. We also plan to test the role of this molecule in life span. As per our response to the comment for reviewer #1, analyses of the Reference Expression Dataset (RefEx), Tabula Muris Senis (<https://tabula-muris-senis.ds.czbiohub.org>) and our qPCR data suggest that PCPE-1 is predominantly produced from BAT in mice at least. As pointed out by the reviewer, the fundamental question here is “why this needs to be predominantly produced from BAT?”, which we find an interesting question to be explored in our future studies. We have therefore added the following statement in the Discussion section: “Finally, a fundamental question that remains to be answered is why this molecule is predominantly produced from BAT. Thus, further studies are required to comprehensively understand the role of this molecule under both physiological and pathological conditions.”

Referee #2:

In this study, the authors identified procollagen C-endopeptidase enhancer (PCPE-1) as a BATokine. The data shows that PCPE-1 is induced in brown adipose tissue (BAT) in response to a NASH model (prolonged high-fat feeding) and is secreted as BATokine, promoting fibrosis in mice. Either BAT-specific or global deletion of PCPE-1 prevented liver fibrosis in this model. In

contrast, PCPE-1 overexpression in BAT was sufficient to promote liver fibrosis in chow-fed mice. The authors propose that HFD-induced ER stress in BAT leads to the activation of IRE1/JNK/c-Fos/c-Jun signaling, thereby inducing PCPE-1. The idea that PCPE-1 is induced in the context of liver fibrosis and may contribute to its development is not new, but this is the first time PCPE-1 is reported as a BATokine. The data suggest that BAT-liver communication mediated by PCPE-1 is critical for obesity-induced liver fibrosis. Although of interest, there are concerns regarding the data presented that impact the conclusions that can be drawn. For example, questions remain regarding the mechanisms mediating PCPE-1 induction in BAT, and regarding whether liver PCPE-1 levels itself also contribute to the development of fibrosis.

Response:

We thank the reviewer for these valuable comments and questions. Please see our response to the “Specific comments” in the following section.

Finally, although PCPE-1 could contribute to the development of liver fibrosis, other aspects of NASH were not ameliorated by PCPE-1 deletion, reducing the significance of these findings as a therapeutic strategy for the treatment of NASH.

Response:

The reviewer’s point is well taken. As described in the review recently published by Harrison et al. (Nature Med. 2023: <https://doi.org/10.1038/s41591-023-02242-6>), there are currently no approved licensed drugs for MASH, and while MASH patients with advanced fibrosis are at high risk for morbidity and mortality from the disease, however, at this stage, there are no robust data showing the contribution of antifibrotic therapy to the hard clinical outcome of patients with MASH. As described in this review and as per the reviewer’s comments, we agree that combination therapy may be the way forward in treating MASH, and as Harrison et al. noted, targets for MASH therapy may be directed at: 1) energy imbalance; 2) inflammation; and 3) fibrosis, and we believe our studies may be of interest in that they indicate PCPE-1 among the interesting molecules to target in suppressing fibrosis in the liver. Thus, we have now included the following statement in the Discussion section: “Multiple mechanisms are involved in the pathogenesis of MASH and include excessive lipid influx and chronic sterile inflammation in the liver. Again, while MASH patients with advanced fibrosis are at high risk for morbidity and mortality from the disease; however, at this stage, there are no robust data showing the benefit of

antifibrotic therapy in the clinical outcome of patients with MASH. Thus, the combination therapy approach may be critical for the treatment of MASH, including modulators of the energy imbalance, inflammation, and fibrosis.”

Specific Comments

1) Although prolonged HFD induced high PCPE-1 levels in BAT, it also significantly induced PCPE-1 levels in the liver. Therefore, it is possible that liver PCPE-1 induction is sufficient to promote fibrosis in this obesity-induced NASH model. Several questions remain to be addressed: 1. Are PCPE-1 levels in liver affected by PCPE-1 deletion and/or overexpression in BAT? 2. Is liver PCPE-1 induction also required to mediate fibrosis in this model? 3. Would conditional deletion of PCPE-1 in the liver result in a similar phenotype as in BAT KO mice? Also, would overexpressing PCPE-1 in the liver be sufficient to induce fibrosis?

Response:

We thank the reviewer for these comments and questions. In response, we analyzed the expression of PCPE-1 in the liver of AAV-mPcolce mice and demonstrated an increased level of this protein in the AAV group, with this result included as a new figure (Fig. EV2P).

We have also shown that PCPE-1 expression was increased in the liver in mice with dietary obesity but was reduced in the BAT KO mice, with these results included as new figures given below at the next page (Figs. 1M and 2D).

Regarding the role of PCPE-1 produced in non-BAT organs, similar questions were raised by reviewer #1. Please see our response for the comment 2) by reviewer #1. Briefly, our analysis of the RefEx (<https://refex.dbcls.jp/index.php?lang=en>)(<https://www.nature.com/articles/sdata2017105>) demonstrated that the expression of *Pcolce* was much lower in the liver than in the adipose tissue. We also tested hepatocyte (AML12) and brown adipocyte cell lines and demonstrated that *Pcolce* was about 500-fold more highly expressed in the brown adipocytes (see the new figure, Fig. EV1P). Furthermore, analysis of primary hepatocytes and brown adipocytes also demonstrated that the level of expression of *Pcolce* was more than 10,000-fold higher in primary brown adipocytes (see the new figure, Fig. EV1O). As per our response to reviewer #1, our analysis of single-cell RNAseq data in the Tabula Muris Senis (<https://tabula-muris-senis.ds.czbiohub.org>) also demonstrated the expression of *Pcolce* in BAT, WAT, aorta and heart (see the new figure, Fig. EV1K-M). Again, analysis of GSE8044 at NCBI-GEO demonstrated that the transcript *Pcolce* was about 2.5-3-fold more highly expressed in BAT than in WAT (see the new figure, Fig. EV1D). The Tabula Muris also showed fibroblasts to be a predominant source of *Pcolce* in the heart (see the new figure, Fig. EV1L). In contrast, analysis of mesenchymal stem cells of adipose tissue showed the highest profile for this gene in BAT (see the new figure, Fig. EV1M). We also performed additional experiments with primary fibroblasts and brown adipocytes collected from neonatal rats and demonstrated that the transcript *Pcolce* was 2.5-3-fold more highly expressed in brown adipocytes (see the new figure, Fig. EV1N). As pointed out by the reviewer, PCPE-1 production from the liver may have some role, but based on these results we speculate that it has only a minor role. While, due to the time constraints suggested by the Editor, we were unable to generate and analyze AlbuminCre^{+/-}; *Pcolce*^{flox/flox} and/or liver-specific *Pcolce*-overexpressing mice as suggested, now we have the following statement to include in the discussion section:

“Our quantitative PCR study and analysis of GSE123394 showed that PCPE-1 is also produced from the liver and increased with dietary obesity. While at this stage we were unable to generate and test a liver-specific PCPE-1 depletion model, which is a limitation of our study, transcriptome studies from *in vivo* and *in vitro* samples indicate that the contribution of liver-derived PCPE-1 may be rather small.”

2) High PCPE-1 levels have been associated with liver fibrosis in previous studies and have been suggested as a potential biomarker and therapeutic target (PMCID: PMC49614444; PMC8377038; PMC5602882). Indeed, global PCPE-1 KO reduced liver fibrosis, but not NASH progression in mice (PMC8836302). The present study should be discussed in light of these initial observations. These studies should be cited and discussed in the present manuscript.

Response:

We are grateful for this suggestion. In response, we have added the following statement in the Discussion section: “Studies in rodents and humans showed that circulating PCPE-1 is associated with liver fibrosis (Hassoun et al, 2017; Hassoun et al, 2016) and systemic depletion of PCPE-1 ameliorates fibrosis in the liver in a murine MASH model (Sansilvestri Morel et al, 2022), indicating that PCPE-1 may represent a relevant biomarker as well as a therapeutic target for MASH (Lagoutte *et al.*, 2021).”

3) Serum levels of PCPE-1 are highly variable in the different settings described. In figure 1F PCPE1 levels in NC controls is 100ng/ml rising to ~ 275ng/ml with HFD. Levels in the AAV studies are 10ng/ml in the mock rising to 25 ng/ml in the AAV. These levels are lower than the normal range in figure 1, yet authors are invoking that this is sufficient to induce fibrosis. This is

somewhat confusing and not consistent with a dose-response effect. Moreover, circulating levels of PCPE1 in humans are 2ng/ml. How much circulating PCPE-1 is required to elicit liver fibrosis? Why are these levels so variable? Does feeding status or any metabolic changes also affect PCPE-1 circulating levels?

Response:

We thank the reviewer for raising these issues. To address these comments, we studied plasma samples with the new ELISA kit and determined the PCPE-1 concentration as around 5 ng/mL in mice fed NC and 10 ng/mL in mice fed HFD (see figures shown below). In response to comment 11), we developed a HFD-fed, AAV *mPcolce* model as well as an NC-fed model and compared these models using the new ELISA kit. Our findings on circulating PCPE-1 levels suggest that PCPE-1 would become pathogenic in mice at 8-10 ng/mL. Incidentally, it was also shown in a cohort of healthy individuals (n = 200) currently being studied that PCPE-1 had an average circulating level of 0.61 ng/mL in humans, suggesting that the PCPE-1 levels appear to vary between different species.

We are also interested in testing whether feeding status or metabolic change may affect PCPE-1 levels. Our ongoing study indicates that BAT thermogenesis is reactivated in HFD-fed mice when subjected to exercise and switched to a normal chow diet and that this is associated with a reduction in BAT and circulatory PCPE-1 levels. Given that these results are still too preliminary,

however, we hope to be able to report more robust results in a separate manuscript in the near future.

4) *There is an overall lack of characterization on BAT function and morphology in BAT PCPE-1 KO mice or in the overexpression model. Does BAT-derived PCPE-1 play any autocrine or paracrine roles that could potentially contribute to the liver phenotype?*

Response:

We thank the reviewer for these insightful comments. To address the reviewer's comments, we performed additional analyses and demonstrated the results of HE staining for BAT in all mice studied as being similar across all genotypes and groups studied (see new figures shown below, i.e., Figs. EV2F, EV2M, EV3G, EV3O, Appendix Figs. S2A, S2C, S2D, S4A, and S4C), with the UCP-1 levels compared in BAT of Con vs BAT Pcolce KO mice and Mock vs AAV-mPcolce mice shown to be similar. We also found that administration of a recombinant PCPE-1 protein to brown adipocytes led to no change in their UCP-1 levels.

Given that, due to the relocation of our lab last April, during our manuscript revision, only systemic PCPE-1 KO mice were available for testing for BAT function, i.e., catecholamine-induced thermogenesis in the interscapular area (doi.org/10.1016/j.molmet.2014.04.007), and we found that administration of CL-316,243 (0.033 nmol/g) led to no marked increase in temperature of the interscapular region in both genotypes subjected to dietary obesity, as shown below.

5) Although body weight was unchanged between WT and PCPE-1 BAT KO mice fed HFD, it is

unclear whether any other metabolic parameters were affected by PCPE-1 manipulation such as glucose homeostasis and body composition.

Response:

We thank the reviewer for raising this point. To address the reviewer’s comments, we have compared epididymal WAT and inguinal WAT in the BAT Pcolce KO mice and found that they were comparable in weight across the genotypes studied. While, as noted above, due to the relocation of our lab last April, at this time, only systemic PCPE-1 KO were made available for testing for glucose tolerance test (GTT), insulin tolerance test (ITT) and CT scan, we noted no change in these tests across the genotypes compared and have shown these results in new figures, i.e., Figs. EV3H, EV3I, EV3J as wells Appendix Figs. S2B and S4B.

6) Several other genes related to fibrosis were identified to be induced in BAT in response to HFD, including *Cyr61*, which has anti-fibrotic properties. Is the expression of these genes altered upon PCPE-1 manipulation in BAT?

Response:

We are grateful to the reviewer for raising this point. We have now performed additional studies as suggested. In the transcriptome data (GSE28440), we found that profibrotic genes including *Adamts2*, *Ccn2* (*Ctgf*), *Tgfb1* increased in dietary-obese BAT. Together with the anti-fibrotic gene, *Ccn1* (*Cyr61*), we tested whether PCPE-1 manipulation may alter the expression profiles of these genes and found that PCPE-1 manipulation did not alter their expression levels in the AAV-mPcolce and BAT Pcolce KO mice (see the new figures below, i.e., Appendix Figures S2E-G).

Appendix Figure S2E

Gene Symbol	Relative mRNA (DIO/Con)	p value
Adamts2	1.4	p<0.01
Ccn2 (Ctgf)	1.3	p<0.05
Tgfb1	1.2	p=0.06

Appendix Figure S2F

Appendix Figure S2G

7) Although AAV-mediated activation of *c-Fos* and *c-Jun* is sufficient to induce PCPE-1 in brown adipocytes, the link with ER stress and IRE1 signaling and the requirement of *c-Fos* and *c-Jun* for PCPE-1 induction were not rigorously tested. Therefore, regarding the mechanisms for PCPE-1 induction in BAT, there are several issues that need to be addressed:

Response:

Thank you very much for these comments. We would like to note that similar questions were also made by reviewer #1 and that we have performed additional studies to address these issues as shown below.

1. Does inhibition of IRE1 and JNK prevent PCPE-1 induction in and secretion from brown adipocytes?

Response:

Our additional studies demonstrated that the inhibition of IRE1 and JNK led to a reduction in BRE-induced Pcolce induction (see the new figure below, Fig. 4L).

Fig 4L

The BRE treatment increased PCPE-1 in the conditioned medium, which we found was suppressed with the inhibition of IRE1 and JNK (see the new figure below, Fig. 4M).

Fig. 4M

2. Does deletion of c-Fos or c-Jun prevent PCPE-1 induction?

Response:

Based on our additional studies, it was confirmed that inhibition of c-Fos/activator protein (AP)-1 (T5224) led to a reduction in BRE-induced Pcolce induction (New figure 4N).

Fig. 4N

3. Does brefeldin (BRE) treatment induce PCPE-1?

Response:

BRE treatment of brown adipocytes led to an increase in the transcript *Pcolce* (see the new figure below, Fig. EV4G).

4. Can induction of PCPE-1 in BAT of obese animals or in brown adipocytes in vitro, be prevented by global attenuation of ER stress with a chemical chaperone, such as TUDCA?

Response:

We confirmed that treatment of brown adipocytes with TUDCA led to a reduction in BRE-induced *Pcolce* induction (see the new figure below, Fig. 4K).

We also examined the reagent TUDCA for its effect on BAT in obese animals and found that it led to a reduction in the transcript *Pcolce* (see the new figure below at next page, Fig. 4O).

Fig. 4O

5. Would activation of ER stress with more classical compounds such as tunicamycin and thapsigargin promote PCPE-1 induction in brown adipocytes?

Response:

We found that treatment of brown adipocytes with the reagents tunicamycin and thapsigargin led to an increase in the transcript *Pcolce* (see the new figures below, Figs. EV4H and EV4I).

6. The investigators have not demonstrated *in vivo* that inhibiting ER stress is sufficient to prevent the secretion of PCPE1. This could be achieved by treating animals with the chemical chaperone TUDCA.

Response:

We are grateful for this helpful suggestion. We have conducted further experiments and found

that TUDCA led to a reduction in circulatory levels of PCPE-1 in mice with dietary obesity (see the new figure below, Fig. 4P).

8) Many of the data panels lack requisite controls. See for example Figures 2e, and all data panels in figure 3 should include data from normal chow-fed controls.

Response:

We thank the reviewer for these comments. As suggested, we examined liver collagen 1 in normal chow-fed control mice and included all these results as a 4 group-study panel for the systemic KO group in Figure 3. We have also conducted additional studies, including CT scans, glucose or insulin tolerance tests, and the results of these are now shown in new figures (Fig. 2F, Figs. 3A-I, EV2H, and EV3A-J). Again, while we had intended to test PCPE-1 induction for its therapeutic potential in the MASH model as well, due to the specified deadline for revision, however, we were unable to do so by incorporating lean mice as controls.

Figure 3

9) It is curious that no plasma concentrations of PCPE-1 are shown in control and HFD mice in the germline KO experiments depicted in Figure 3.

Response:

We thank the reviewer for these helpful comments. We have conducted further ELISA studies and found a reduction in PCPE-1 in the systemic Pcolce KO mice as shown in the new figure below, Fig. 3C.

10) It is intriguing that BAT Pcolce KO mice also have reduced TG. Why is this? And how does this contribute to the amelioration of the fibrosis phenotype?

Response:

Please see our response to comment 11.

11) Would HFD exacerbate the findings in the AAV mouse? It seems like the degree of fibrosis observed under in these mice under NC conditions is significantly less. They do seem to have a slightly more hepatic TG although there is no inflammation. This again raises the question regarding whether or not PCPE-1 could have pleiotropic effects.

Response:

We thank the reviewer for these comments. We include our responses to comments 10 and 11 here.

As suggested, we conducted additional studies and found that HFD exacerbates fibrosis in the liver in our AAV model (see the new figures below, Fig. 2N, Fig. EV2Q and Appendix Figs. S2H and S2I). Of the findings obtained, that on liver TG levels appears to be of particular interest: as

already demonstrated, with BAT-specific PCPE-1 depletion, TG was decreased in mice with dietary obesity, but tended to be increased in the normal chow-fed AAV-mPcolce model and was significantly increased in the HFD AAV-mPcolce model (see the new figure below, Fig. EV2R).

To further test the potential role of PCPE-1 in lipid metabolism, we treated primary hepatocytes while under steatotic conditions (doi: 10.1016/j.jhep.2013.07.019.Epub 2013 Jul 19.) with a recombinant PCPE-1 protein. PCPE-1 induction did not alter the expression profiles of the genes tested, except for *Ldlr* which showed a reduction in the steatosis group (see the new figure below, Appendix Fig. S3A-D). It was also shown that the TG levels increased in steatotic hepatocytes but with this change not affected by PCPE-1 induction (see the new figure below, Appendix Fig. S3E).

Appendix Figure S3

As the reviewer noted, our data indicate the potential pleiotropic effect of PCPE-1 and/or fibrosis in lipid metabolism. We hope to explore this potential in our future studies and have included the following statement in the Discussion session: “Interestingly, liver TG levels were shown to be significantly reduced in our *BAT-Pcolce* KO model with dietary obesity, while *Pcolce* overexpression in *BAT* tended to increase liver TG levels in our AAV-*mPcolce*-induced gain-of-function model and significantly increased liver TG levels under obese conditions. To further test PCPE-1 for its potential role in lipid metabolism, primary hepatocytes were treated, under steatotic conditions with a recombinant PCPE-1 protein. PCPE-1 treatment did not alter the expression profiles of the genes tested, except for *Ldlr*, which was shown to be reduced in the steatosis group. Again, intracellular TG levels were increased in steatotic hepatocytes, while this change was not affected by the PCPE-1 treatment. Our data indicate the potential pleiotropic effect of PCPE-1 and/or fibrosis in lipid metabolism; however, further studies are needed to test this hypothesis.”

Comments on the organization of the manuscript

1. As written the manuscript is not easy to follow. The authors include too much background and discussion in the presentation of the results. Some of these points should either be briefly consolidated in the introduction or moved to the discussion section.

Response:

As suggested, we have now revised our manuscript to improve readability. We hope you will find the revised manuscript more readable and easier to follow.

2. The authors might consider reorganizing the flow of the data by initially focusing on the regulation of PCPE-1 in BAT prior to describing its putative role in the regulation of hepatic fibrosis.

Response:

We are grateful to the reviewer for this kind suggestion. As per the reviewer's suggestion, we have considered reorganizing our data but concluded that the current flow of argument may be found easily accessible. We will of course be ready to address any concerns this may raise.

3. The manuscript should be edited for grammar and some typos fixed such as Picrosirius in the figures... which is incorrectly labeled as Picro Sillius in figs 1j, 2d,m, 3d,j.

Response:

We apologize for any inconvenience caused. As per the reviewer's suggestion, we have had our entire manuscript read for grammar, style and consistency, with all changes made highlighted in red, and hope our edited version will now be found in order and acceptable.

4. The terms Non-Alcoholic Steatohepatitis (NASH) and Non-Alcoholic Fatty Liver Disease (NAFLD) have been replaced Metabolic Dysfunction Associated Steatohepatitis (MASH) and Metabolic Dysfunction Associated Fatty Liver Disease (MAFLD). The terminology used in the manuscript should be replaced with this more contemporary nomenclature.

Response:

We thank the reviewer for these helpful comments. We have now replaced the technical terms with those in current use throughout the text.

Dear Dr Shimizu,

Thank you for submitting your revised manuscript (EMBOJ-2023-115949R) to The EMBO Journal. As mentioned, your amended study was sent back to the referees for their re-evaluation, and we have received comments from both of them, which I enclose below. As you will see, the experts stated that the work has been substantially improved by the revisions and they are now broadly in favour of publication.

Thus, we are pleased to inform you that your manuscript has been accepted in principle for publication in The EMBO Journal.

We now need you to take care of a number of minor issues related to formatting and data presentation as detailed below, which should be addressed at re-submission.

Please contact me at any time if you have additional questions related to below points.

As you might have seen on our web page, every paper at the EMBO Journal now includes a 'Synopsis', displayed on the html and freely accessible to all readers. The synopsis includes a 'model' figure as well as 2-5 one-short-sentence bullet points that summarize the article. I would appreciate if you could provide this figure and the bullet points.

Thank you for giving us the chance to consider your manuscript for The EMBO Journal.
I look forward to your final revision.

Again, please contact me at any time if you need any help or have further questions.

Kind regards,

Daniel Klimmeck

>> Please add up to five keywords to your study.

>> Author Contributions: Please remove the author contributions information from the manuscript text. Note that CRediT has replaced the traditional author contributions section as of now because it offers a systematic machine-readable author contributions format that allows for more effective research assessment. and use the free text boxes beneath each contributing author's name to add specific details on the author's contribution.

More information is available in our guide to authors.
<https://www.embopress.org/page/journal/14602075/authorguide>

>> Add a 'Disclosure and Competing Interests Statement' to the manuscript.

>> Section order should be corrected as follows: title page with complete author information, abstract, keywords, introduction,

results, discussion, methods, data availability section, acknowledgements, disclosure and competing interests statement, references, main figure legends, tables, expanded figure legends.'

>> Figures in separate files: Figures should be removed from the manuscript and uploaded as individual, high resolution figure files.

>> Callouts: callouts for Tables and Datasets need to be added to the manuscript.

>> Dataset EV legends: Datasets need to be numbered as Dataset EV1, Dataset EV2... with the appropriate callouts.

>> Appendix file: The appendix file needs to be in PDF format starting with ToC with page numbers on its first page. Appendix figure legends should be removed from manuscript file and inserted below the corresponding figures.

>> Funding: information on funding is incomplete in our online system. only one funder is inserted in eJP: Fusion Oriented Research for Disruptive Science and Technology (JST FOREST Program)(JPMJFR200L) - all the others need to be included in the Acknowledgements section.

>> Data availability section: please enter a Data availability section into the manuscript, stating 'No large-scale data amenable to data repository deposition were generated in this study.' . Please remove the current 'Public Database Studies' and 'Materials & Correspondence' sections and add respective information to the new Data availability section. Indicate the institutional author e-mail address.

>> Please remove the one-sentence summary from the manuscript.

>> Source data: missing numerical data in Dataset for main figures: Fig. 1C, 1F, 1H, 1I, 2A, 2C, 2F, 2G, 2H, 2J, 2K, 2M, 2O, 2P, 2Q, 3A, 3C, 3F, 3G, 3H, 3J, 3K, 3M, 3N, 4C, 4E; please provide a completed source data checklist as to the separate instructions by my colleague Hannah Sonntag. Source data files need to be reorganized to one file/folder per figure and ZIPing for each main figure. For EV and/or appendix figures, ZIP together all source data.

>> Consider additional changes and comments from our production team as indicated below:

- Data Availability Section: Please note that the data availability statement is not provided in the manuscript.

- Figure Legends (main + EV):

1. Please note that a separate 'Data Information' section is required in the legends of figures 1c-f, h-i, k-n; 2a, c-d, f-i, k-m, o-r; 3a, c, f-i; k, m-p; 4a-p; EV 1a-b, d-j, n-q; EV 2b-e, g-l, n-p, r-s; EV 3a-f, i-n; EV 4b-i.
2. Please note that the legends for figures 2c-e are not provided in the sequential manner (legend for figures 2d, e are provided before legend of figure 2c). This needs to be rectified.
3. Please note that the legends for figures 4m-n is not provided in the sequential manner (legend for figures 4n is provided before legend of figure 4m). This needs to be rectified.
4. Please note that the legends for figures EV 3c-e are not provided in the sequential manner (legend for figure EV 3e are provided before legends of figures EV 3c-d). This needs to be rectified.
5. Please define the annotated p values ## / # in the legend of figure EV 3j as appropriate.
6. Please indicate the statistical test used for data analysis in the legend of figure 1a.
7. Although 'n' is provided, please describe the nature of entity for 'n' in the legends of figures 2k; 4e-f, i-j, k-l, n, EV 1d-j, n-p; EV 2b-c, j, EV 3b, e, i, k, EV 4b-i.

- Data citations:

1. Please note that the data callouts in the text for all the data citations does not include "Data ref:" as a prefix.
2. Please note that although dataset specific URL is provided, the URLs are currently invalid.

Referee #1:

Thank you for addressing the reviewers' points in this thoroughly revised manuscript. In future, please refrain from using single lane immunoblots for quantification.

Referee #2:

The authors have performed additional experiments to bolster their conclusion that PCPE1 is a BAT-derived adipokine that increases hepatic fibrosis. They also show that genetically reducing PCPE1 in BAT, reduces fibrosis and similar findings are observed in a germline KO or animals treated with a neutralizing antibody. The authors also provide evidence that an important mechanism for the the induction of PCPE1 in BAT is secondary to activation of ER stress. Although they show that circulating levels of PCPE are increased in humans with NASH, this does not prove that the source in humans in coming from BAT, given that PCPE1 is also increased in the livers in MASH. Thus, overall the authors demonstrate at least in rodents, that BAT derived PCPE1 correlates with increased circulating levels and increased hepatic fibrosis, without reducing hepatitis or steatosis. There is little else that the authors can do to strengthen their conclusions.

Referee #1:

Thank you for addressing the reviewers' points in this thoroughly revised manuscript. In future, please refrain from using single lane immunoblots for quantification.

Response:

We thank the reviewer for these kind suggestions and comments.

Referee #2:

The authors have performed additional experiments to bolster their conclusion that PCPE1 is a BAT-derived adipokine that increases hepatic fibrosis. They also show that genetically reducing PCPE1 in BAT, reduces fibrosis and similar findings are observed in a germline KO or animals treated with a neutralizing antibody. The authors also provide evidence that an important mechanism for the the induction of PCPE1 in BAT is secondary to activation of ER stress. Although they show that circulating levels of PCPE are increased in humans with NASH, this does not prove that the source in humans in coming from BAT, given that PCPE1 is also increased in the livers in MASH. Thus, overall the authors demonstrate at least in rodents, that BAT derived PCPE1 correlates with increased circulating levels and increased hepatic fibrosis, without reducing hepatitis of steatosis. There is little else that the authors can do to strengthen their conclusions.

Response:

Thank you very much for these comments. We totally agree with the reviewer's comments regarding the source of PCPE-1 in humans. In the discussion section, we added "in mice" for the following section.

Combined results from quantitative PCR studies and the BAT-specific Pcolce KO mice provide evidence that PCPE-1 is an adipokine produced predominantly from BAT in mice.

Please also see the description regarding the limitation of our paper in the same section. Thank you very much.

While the source of this molecule in humans remains unclear, this may not raise major issues for drug development, given that PCPE-1 is a secreted protein. Patients with MASH are shown to exhibit increased circulatory levels of PCPE-1, suggesting that the suppression of this pro-fibrotic protein may represent a novel treatment modality for MASH.

Dear Dr Ippei Shimizu,

Thank you for submitting the revised version of your manuscript. I have now evaluated your amended manuscript and concluded that the remaining minor concerns have been sufficiently addressed.

I am thus pleased to inform you that your manuscript has been accepted for publication in the EMBO Journal.

Also, we kindly ask for your consent on keeping the referee figures included in this file.

On a different note, I would like to alert you that EMBO Press offers a format for a video-synopsis of work published with us, which essentially is a short, author-generated film explaining the core findings in hand drawings, and, as we believe, can be very useful to increase visibility of the work. Please see the following link for representative examples and their integration into the article web page:

<https://www.embopress.org/doi/full/10.15252/emj.2019103932>

If you have any questions, please do not hesitate to contact the Editorial Office.

Best regards,

Daniel Klimmeck

Daniel Klimmeck, PhD
Senior Editor
The EMBO Journal
EMBO
Postfach 1022-40
Meyerhofstrasse 1
D-69117 Heidelberg
contact@embojournal.org